# Revisiting silk: a lens-free optical physical unclonable function

Min Seok Kim [1,8], Gil Ju Lee[1,2,8], Jung Woo Leem[3], Seungho Choi[4], Young L. Kim [3,5 ✉] & Young Min Song [1,6,7 ✉]

For modern security, devices, individuals, and communications require unprecedentedly unique identifiers and cryptographic keys. One emerging method for guaranteeing digital security is to take advantage of a physical unclonable function. Surprisingly, native silk, which has been commonly utilized in everyday life as textiles, can be applied as a unique tag material, thereby removing the necessary apparatus for optical physical unclonable functions, such as an objective lens or a coherent light source. Randomly distributed fibers in silk generate spatially chaotic diffractions, forming self-focused spots on the millimeter scale. The silk-based physical unclonable function has a self-focusing, low-cost, and eco-friendly feature without relying on pre-/post-process for security tag creation. Using these properties, we implement a lens-free, optical, and portable physical unclonable function with silk identification cards and study its characteristics and reliability in a systemic manner. We further demonstrate the feasibility of the physical unclonable functions in two modes: authentication and data encryption.

[1] School of Electrical Engineering and Computer Science (EECS), Gwangju Institute of Science and Technology, 123, Cheomdangwagi-ro, Buk-gu, Gwangju 61005, Republic of Korea. [2] Department of Electronics Engineering, Pusan National University, 2 Busandaehakro 63 beon-gil, Geumjeong-gu, Busan 46241, Republic of Korea. [3] Weldon School of Biomedical Engineering, Purdue University, West Lafayette, IN 47907, USA. [4] Department of Biomedical Engineering, Yonsei University, Wonju 220-710, Republic of Korea. [5] Purdue Quantum Science and Engineering Institute, West Lafayette, IN 47907, USA. [6] Anti-Viral Research Center, Gwangju Institute of Science and Technology (GIST), 123, Cheomdangwagi-ro, Bukgu, Gwangju 61005, Republic of Korea. [7] AI Graduate School, Gwangju Institute of Science and Technology (GIST), 123, Cheomdangwagi-ro, Bukgu, Gwangju 61005, Republic of Korea. [8]These authors contributed equally: Min Seok Kim, Gil Ju Lee. ✉email: youngkim@purdue.edu; ymsong@gist.ac.kr

Counterfeit consumer products have led to billion-dollar economic losses; in particular, fake medical devices/medicines have been threatening human safety and public health in both developed and less developed countries[1–5]. Moreover, in the digital (or information) age, confidential and private information has been significantly threatened for the mercenary and/or abusive purposes of cyber terrorism in every area of modern society, such as communication, military, finance, and personal data[6–8]. To abate such counterfeit goods and attacks on privacy/classified information, authentication is an essential cryptographic primitive that validates the identifier for authorized products and users[9]. As relatively unbreakable identifiers, physical(ly) unclonable functions (PUFs), which are non-algorithmic one-way functions composed of uncopiable elements, have been extensively highlighted[10,11]. Because PUFs result from a random or stochastic process, PUF-based tags pose a truly unique feature once the tag has been manufactured.

After the initial introduction of PUF concepts, various types of PUFs using optics[12–16], magnetics[17–23], electronics[24–28], and radio frequency[29,30] have been reported. Compared to other PUFs, optical-based PUFs have the advantages of high entropy, high output complexity, and high security against modeling and cloning attacks[15]. However, optical PUFs require bulk readout systems equipped with multiple lenses or objective lenses for fine-tuning the focus. Hence, there remains much room for improvement, such as reducing the bulkiness, cost, and complexity (Supplementary Table 1).

Herein, we introduce a lens-free optical PUF system based on stochastically manifested diffraction using native silk fibers. Silk produced by silkworms (i.e., *Bombyx mori*) has been extensively utilized as a fabric and biomaterial due to its various merits of superior mechanical properties, biocompatibility, and biodegradability[31–33]. Recently, silk has received attention as a versatile photonic metamaterial since researchers revealed the photonic phenomenon, 'Anderson light localization', in silk fibers due to its randomly distributed nanofibrillar structures in microfibers[34,35]. In addition to these nanostructures, however, we highlight the inherently chaotic arrangement of silk microfibers for an optical PUF application. Our theoretical and experimental analyses reveal two intriguing features for a lens-less optical PUF: (1) the proper microfiber density exhibits a 'self-focusing' phenomenon by diffraction from arbitrarily formed pinholes and (2) the nanostructures in individual microfibers enhance a light intensity contrast with respect to the background. In particular, these features indicate that intense focal spots can be formed within the region of Fraunhofer diffraction, which can be observed on the millimeter scale. In addition, diffraction-based self-focusing can be performed using affordable incoherent light sources, such as light-emitting diodes (LEDs).

The silk fiber-based optical PUF not only has extraordinary optical characteristics, but is also low-cost, eco-friendly, without requiring pre-/post-processing for PUF-tag creation. Based on these features, we implement a lens-free, optical, and portable PUF (LOP-PUF) module by optimizing the distance between a silk PUF-tag and an image sensor. This simple apparatus easily forms random light-spot patterns with a high-intensity contrast. Hence, the proposed module significantly reduces the complexity, bulkiness, and cost of module production. To show its applicability and practicality, we evaluate the excellent PUF features of the LOP-PUF, as follows: (1) readout system reliability (i.e., humidity, thermal noise, and aging effect); (2) a bit uniformity of ~0.5; (3) a readout reproducibility (i.e., an intra-device Hamming distance (HD)) of ~0.03; (4) a uniqueness (i.e., an inter-device HD) of ~0.5; and (5) a randomness that successfully passes the NIST test. Using the proposed module, we finally demonstrate the feasiblity of the LOP-PUF for strong authentication, which is difficult to break in a polynomial time, and data encryption presenting reliably encrypted and restored data.

## Results and discussion

**Stochastic random holes for 'self-focusing' in native silk**. Native silk, which is a disordered fibrous biomaterial produced by silkworms, exhibits stochastic random holes in space depending on its density (Fig. 1a). An optimal density of disordered fibers allows the transmitted light through holes in silk mcirofibers to strongly focus on the image plane, being captured as light spots with high intensity by an image sensor. This self-focusing effect is attributable to optical diffraction at holes with moderate sizes. At a higher density of silk fibers, the size and number of holes significantly decreased, blocking the incident light. Lower density of silk fibers leads to large opening holes; thus, strongly focused transmitted light is not formed. In this context, at lower or higher density of silk fibers, the desired focal spots of transmitted light are not generated on an image plane. Fortunately, a silk cocoon consists of hierarchical structures, including abundant nanofibrils within microfibers. This multi-scale geometry provides strong light scattering, which can block the incident light (transmission suppression), improving the light intensity contrast between the focal spots and the background (Fig. 1b). From this scheme, the stochastically randomly distributed holes produced by silk fibers with an optimal density could form a high contrast and unique focused light spots, which can be utilized as random seeds for PUF devices.

The self-focusing effect depends on three factors: the wavelength ($\lambda$) of the incident light, the largest width ($w$) of the holes, and the distance ($d_{IS}$) between the image plane and native silk corresponding to the propagation depth of the light (Fig. 1c) as the propagation of light through the hole resulted in light diffraction. In particular, the diffracted light is concentrated into the center of the hole at a depth of propagation greater than $w^2/\lambda$, which is called the Fraunhofer region. The width of the holes defines the depth position of the concentrated light at a specific wavelength. Figure 1d shows the experimental support of the light concentration through random opening holes within the Fraunhofer region for $d_{IS}$ (Supplementary Fig. 1). Moreover, owing to the interference of diffracted light, the focal spots corresponding to different incident angles are distinctly constructed on the image plane. This feature is a virtue for the application of optical PUFs that require diverse challenge–response pairs. By using spectral multiplexing (i.e., 467, 525, and 637 nm) at different incident angles (i.e., −15°, 0°, and +15°) of light as multiple challenges, the spot points (i.e., responses) could further be formed on the image sensor.

Figure 1e depicts the bit extraction flow of the optical PUF based on the self-focusing feature of native silk. From the raw data, three unique planes are detected under light illumination of three colors (i.e., red, green, and blue) at different incident angles (i.e., −15°, 0° and +15°) (Fig. 1e; (i)). For the removal of noise from the image, a threshold is applied to distinguish peak points (Fig. 1e; (ii)). The binning method is further utilized for the reproducibility of the peak points (Fig. 1e; (iii)). The noise-reduced and resized image is used to generate a bitstream, which serves as a seed for a fuzzy bit extractor (Fig. 1e; (iv)). Finally, the digitized bitstream passes through a von Neumann debiasing process (e.g., fuzzy bit extractor), resulting in improved bit uniformity (Fig. 1e; (v)).

**Analysis of the self-focusing property using the modeling of fibrous media**. To investigate the influence of the fiber density and $d_{IS}$, light propagation is simulated using the beam propagation method in a two-dimensional single slit (Fig. 2a). When the

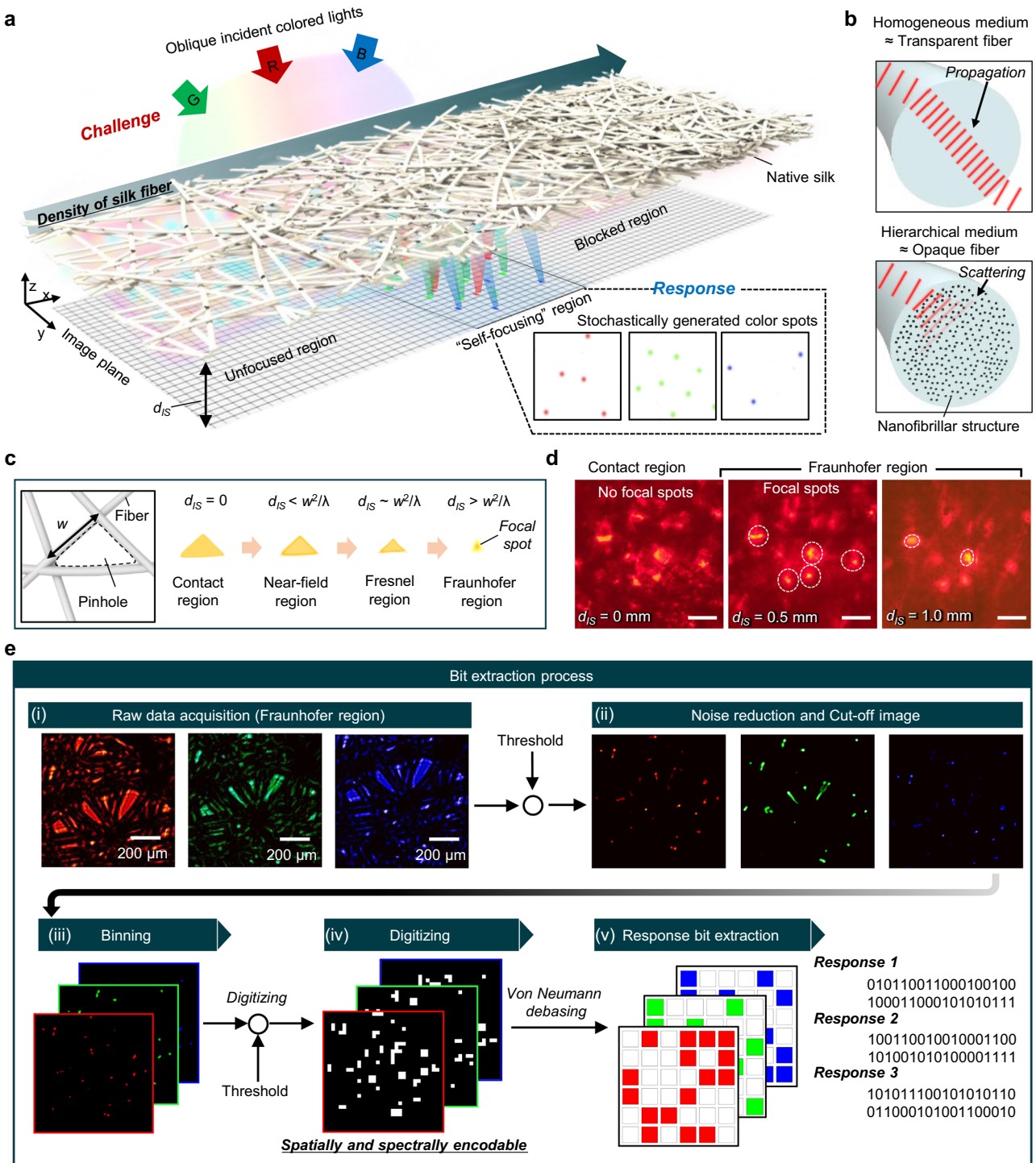

**Fig. 1 Native silk features: self-focusing and unique bit extraction. a** Schematic illustration of native silk with a 'self-focusing' feature caused by diffraction. The diffraction manifested depending on the microfiber density; low and high density of fibers could not produce a self-focused feature due to too large or small pinhole sizes. However, the 'self-focusing' region creates stochastic and unique color spots on an image plane. **b** Schematic illustration for a (top) transparent fiber and (bottom) opaque fiber due to the nanofibrillar structures found in native silk. **c** Operating principle of the self-focusing effect. Randomly distributed fibers form a stochastic pinhole, which causes diffraction. '$d_{IS}$', '$w$', and '$\lambda$' are the distance between the fiber pinhole and image plane, the width of the fiber pinhole, and the incident wavelength, respectively. In the Fraunhofer region (i.e., $d_{IS} > w^2/\lambda$), the diffracted light is self-focused. **d** Obtained image with a red LED at three distances (0, 0.5, and 1.0 mm). The scale bar is 100 μm. **e** Description of the bit extraction process, involving: (i) raw data acquisition, (ii) noise reduction and cut-off image generation, (iii) binning, (iv) digitization, and (v) response bit extraction.

light encounters a fiber, the light experiences diffraction. The interference of the diffracted light creates a self-focusing region, i.e., $Z_{focused}$, defined from $Spot_{start}$ to $Spot_{end}$. The simulated result shows that the fiber with an opening diameter ($D$) of 35 μm creates a wide spot region from 0.3 to 0.9 mm at a wavelength of 645 nm. The maximum intensity of the focal spot (i.e., $Spot_{max}$) appears at a $Z$ position of 0.45 mm. This long spot region allows the removal of a lens element to form strong light spots on an

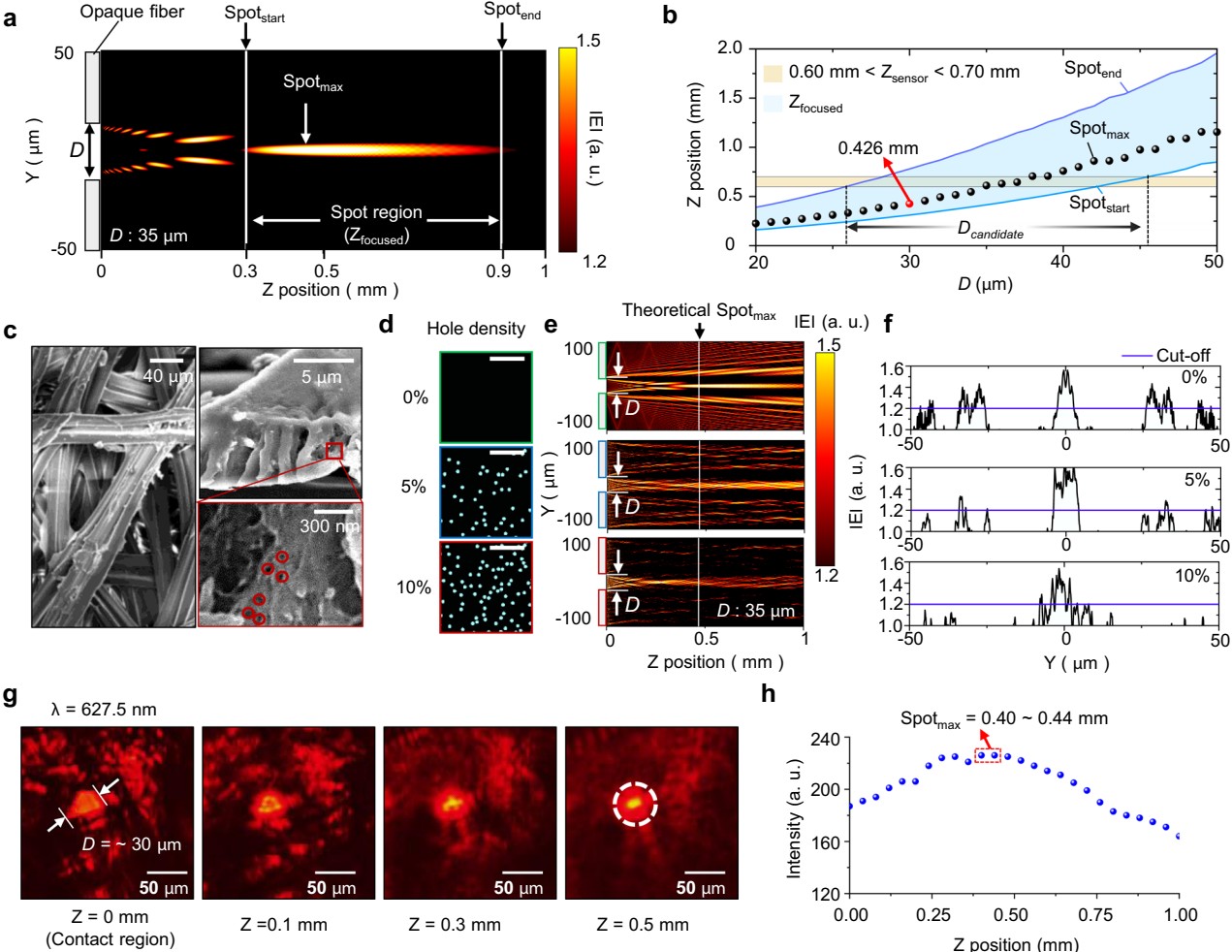

**Fig. 2 Theoretical and experimental analyses on the self-focusing of fibrous media. a** Optical simulation of an opaque fiber with an opening diameter of 35 μm under a wavelength of 645 nm. Fraunhofer diffraction forms an intense spot within a certain spatial range, which is called the 'Spot region'. The 'Spot region' is defined from the $Z$ positions of '$Spot_{start}$' and '$Spot_{end}$'. '$Spot_{max}$' indicates the strongest focal spot. **b** Calculated spot information, including '$Spot_{start}$', '$Spot_{end}$', '$Spot_{max}$', and the 'Spot region', depending on $D$. The 'Spot region' is marked in blue. The active pixel of the image sensor is located at 725 ± 75 μm, which is noted by '$Z_{sensor}$' and highlighted in yellow. The overlapping region of the $Z$ position with the 'Spot region' and '$Z_{sensor}$' is defined as '$D_{candidate}$' which ranges from 27 to 48 μm. **c** Scanning electron microscope images of silk fibers. **d** Refractive index profile of three media with different hole density levels (i.e., 0, 5, and 10%). The black and bluish green colors refer to the refractive indices of 1.5 and 1, respectively. The scale bar is 200 nm. **e** Electric field distributions for three fiber media. **f** Cross-sectional amplitude profiles of three results. **g** Measured data of the silk sample according to the depths (0, 0.1, 0.3, and 0.5 mm). **h** Intensity of the measured data. The maximum intensity of 226 is observed at the $Z$ positions of 0.40 and 0.44 mm.

image sensor. The detailed calculation steps are described in the "Methods" and Supplementary Information.

In general, commercial image sensors require a cover glass and an air gap to protect the active pixels from an undesirable shock. In the image sensor used in this experiment, the thicknesses of the cover glass and the air gap are 0.525 ± 0.05 and 0.125 mm, respectively (Supplementary Fig. 2). By considering the manufacturing tolerance (±0.05 mm), the desired $Z$ position of the sensor (i.e., $Z_{sensor}$) is 0.6–0.7 mm. Figure 2b exhibits the focusing features, i.e., $Spot_{start}$, $Spot_{end}$, $Spot_{max}$, and $Z_{focused}$, depending on $D$. The overlapping area of $Z_{focused}$ and $Z_{sensor}$ provides a candidate range of $D$ (i.e., $D_{candidate}$) for lens-free imaging, where the range of $D_{candidate}$ is from 27 to 48 μm.

The simulations performed in Fig. 2a, b are conducted by considering an ideal case with a totally opaque fiber. Normally, microfibers in paper or textiles consist of visibly transparent materials, such as cellulose or polymers. Native silk microfibers contain nanofibers that form nanoholes (nanoscale void areas), resulting in strong light scattering (or transmission suppression)

(Fig. 2c). Figure 2d shows virtual microfibers with different hole density, such as 0, 5, and 10%. Electric field simulation results on fibers with nanoholes exhibit the ability of the nanofibers to eliminate the side lobe (Fig. 2e). The specific geometry of the simulated fibers is shown in Supplementary Fig. 3. Purely clear microfibers suffer from a low contrast due to intense light at the side; in contrast, microfibers with higher nanohole density present a remarkably improved contrast for self-focusing (Fig. 2f). From this perspective, a native silk cocoon produced by silkworms is a powerful material for optical PUFs.

To validate the theoretical analyses of the self-focusing effect, a customized measurement setup is established to obtain images within the region of $Z_{focused}$, as shown in Supplementary Fig. 4. Figure 2g presents magnified images as a function of the $Z$ position. The original images are displayed in Supplementary Fig. 5. The image at the $Z$ position of 0.1 mm shows the diffraction pattern found in the Fresnel region. At a $Z$ position of 0.5 mm, however, the acquired image exhibits a bright light spot with a size of ~15 μm. In addition, the maximum intensity in each

image is measured depending on the $Z$ position to evaluate the focal features (Fig. 2h). In the case of $D = \sim30\,\mu m$, the maximum intensity (i.e., 226) is observed at $Z$ positions from 0.40 to 0.44 mm; the simulation result also indicates a similar $Spot_{max}$ at $Z = 0.426$ mm (Fig. 2b; red dot).

**Integrated lens-less optical PUF system**. Based on the theoretical analyses above, the LOP-PUF system, which can be a class of optical PUF module, is proposed (Fig. 3a). Generally, an optical PUF system requires bulk readout systems (i.e., a magnification lens, zoom lens, and an extra light source). However, the self-focusing effect-based optical PUF system reduces the bulkiness and additional optical components required for readout. The proposed module consists of a mirror, an image sensor, a field-programmable gate array (FPGA), and three tricolor LEDs. Each LED is located with a 15° angle offset to generate challenge and response pairs. The slanted mirror is placed over an image sensor that redirects light to the image sensor. This structure helps to minimize the size and bulkiness of the optical PUF module. In addition, the native silk is directly located on the image sensor by inserting a silk ID card into the module (Fig. 3b; inset). Using self-focused light spots, the readout system could obtain random seed images without the addition of geometrical optical systems. Figure 3c shows the operating status of the unsealed LOP-PUF.

To investigate the self-focusing feature of the fiber bundles, virtual fiber media are computationally produced with density levels of 70, 80, and 90% (Fig. 3d). The created fiber media have a size of $1 \times 1$ mm$^2$. Figure 3e presents the minimum, maximum, and average diameters (i.e., $D_{min}$, $D_{max}$, and $D_{avg}$) of the opening holes produced by crossing microfibers depending on the density of the random fibrous medium. We repeat the generation of a random fibrous medium. The detailed statistical analyses are shown in Supplementary Fig. 6. As the density increases, the overall size of $D$ decreases; however, $D_{avg}$ presents the optimal density ranges of the fibrous medium to be 60, 70, and 80% for the target image sensor (Fig. 3e). Among them, the fibrous medium with a density of 80% offers the largest $D_{candidate}$ value, implying that this fibrous medium potentially causes a large number of strong light spots within the region of $Z_{sensor}$ (Fig. 3f).

Based on the virtual fibrous media, three-dimensional electric field simulations support this hypothesis (Supplementary Fig. 7). Figure 3g shows the cross-sectional electric field simulation results corresponding to the dashed lines in Fig. 3d. The lower density (i.e., 70%) of silk displays a longer self-focusing distance behind the sensor position. At a higher density (i.e., 90%), the fibrous medium almost blocks light transmission. However, the fibrous medium with the optimal density (i.e., 80%) provids three peaks for the image sensor. A large number of peaks are desirable because the peaks captured by the image sensor are converted to random seeds for the PUF system. From this perspective, too sparse or dense fibrous media offer inappropriate PUF tags for generating high-quality random seeds.

Three native silk samples with different fiber density levels are imaged using the implemented LOP-PUF system (Fig. 3h). As discussed, a silk sample of low density shows a simple shadow of silk fiber, which has an easily predictable pattern (Fig. 3h; (i)). In addition, high density silk causes light blocking due to the absence of effective microholes (Fig. 3h; (iii)). In the case of the proper density, however, the Fraunhofer region matches the distance of the image sensor, which causes bright light spots without optical components (Fig. 3h; (ii)). To binarize the raw image, a threshold is applied to remove 80% of the intensity range of the image (Supplementary Fig. 8). The cut-off images show

that the silk samples with the proper density have dozens of tiny bright spots (Fig. 3h; inset). Moreover, the self-focusing effect maximizes the contrast between the spot and background (Fig. 3i). To test the ability to generate multiple challenges, three LEDs are sequentially activated with different colors (Fig. 3j). The obtained images show different peak positions according to the position of the LED. In addition, the color image sensor could simultaneously record the information of the bright spots from the different incident angles (Fig. 3k). This color space (i.e., ternary bits) exponentially increases the encoding capacity of the PUF module as compared to that of the monochromatic case (i.e., binary bits) (Fig. 3l).

**Reliability of LOP-PUFs**. A high-quality PUF should have high reliability against external and internal factors, such as thermal noise and aging. First, the robustness of the LOP-PUF to thermal noise is estimated by simulating a bit error rate (BER) depending on the signal-to-noise ratio (SNR). Figure 4a illustrates the bit acquisition process to simulate the BER of the LOP-PUF. Ten-bit sequences with a 64-bit size are obtained in one acquisition process, and this process is repeated a hundred times to acquire a thousand bit sequences because the BER test demands a large number of bit sequences. Figure 4b presents the original data as well as the data treated by white Gaussian noise with different intensity levels to satisfy the SNRs of 0, 6, and 12 dB. Other SNRs of treated data (2, 4, 8, 10 dB) are shown in Supplementary Fig. 9. Figure 4c shows the BER of the LOP-PUF versus SNRs. The LOP-PUF shows a low BER ($<10^{-4}$) over the SNR of 10 dB, i.e., when the intensity of signal is 10 times higher than that of noise.

To investigate the actual influence of thermal noise in the LOP-PUF, continuous operation of the LOP-PUF is conducted by measuring the temperature of the module and capturing the responses without and with temperature control (Figs. 4d, e). The obtained responses with and without temperature control are displayed in Supplementary Figs. 10 and 11, respectively. Without the temperature control, the image sensor of LOP-PUF heats up to $\sim35$ °C in 6 min; however, with a cooling fan, the temperature maintains at $\sim27$ °C (Fig. 4d). The LOP-PUF without a cooling fan tends to cause error bits during the operation (Fig. 4e). The LOP-PUF without the cooling fan shows a reduced performance (i.e., # Correct bits/# Total bits) to 90% as the operating time increases (i.e., increase in the temperature of the image sensor). In contrast, the cooling fan allows the LOP-PUF to have a stable performance of $\sim100$%.

In addition, because the LOP-PUF adopts the silk material as the PUF-tag, biodegradability is an important issue as it may reduce the reliability of the LOP-PUF. A customized measurement setup is used to test the biodegradability effect of silk (Fig. 4f). Because silk is susceptible to humidity, our setup could control the humidity using silica gel and a humidifier. The initial relative humidity (RH) is controlled at a low value, i.e., 30.2%, and then it is increased up to 59.7% using the humidifier. While controlling the RH, the LOP-PUF captures the responses and the obtained responses are transformed to bit sequences. The bit extraction is conducted 10 times to evaluate the stability of bit sequences.

The extracted bits at the RH of 30.2, 34.8, 39.8, 44.8, and 50.0% show robust results (i.e., zero-bit error). However, the extracted bits at the RH of 55.2 and 59.7% exhibit 2- and 8-bit errors, respectively (Fig. 4g and Supplementary Fig. 12). During the humidity test, the LOP-PUF exhibits an error rate (i.e., # Error bits/# Total bits) of 0.0022. In addition, a long-term measurement is performed for a week under the conditions of room temperature and humidity. In this result, an error of 1 bit, which corresponds to an error rate of 0.0004, appears at the 6th day

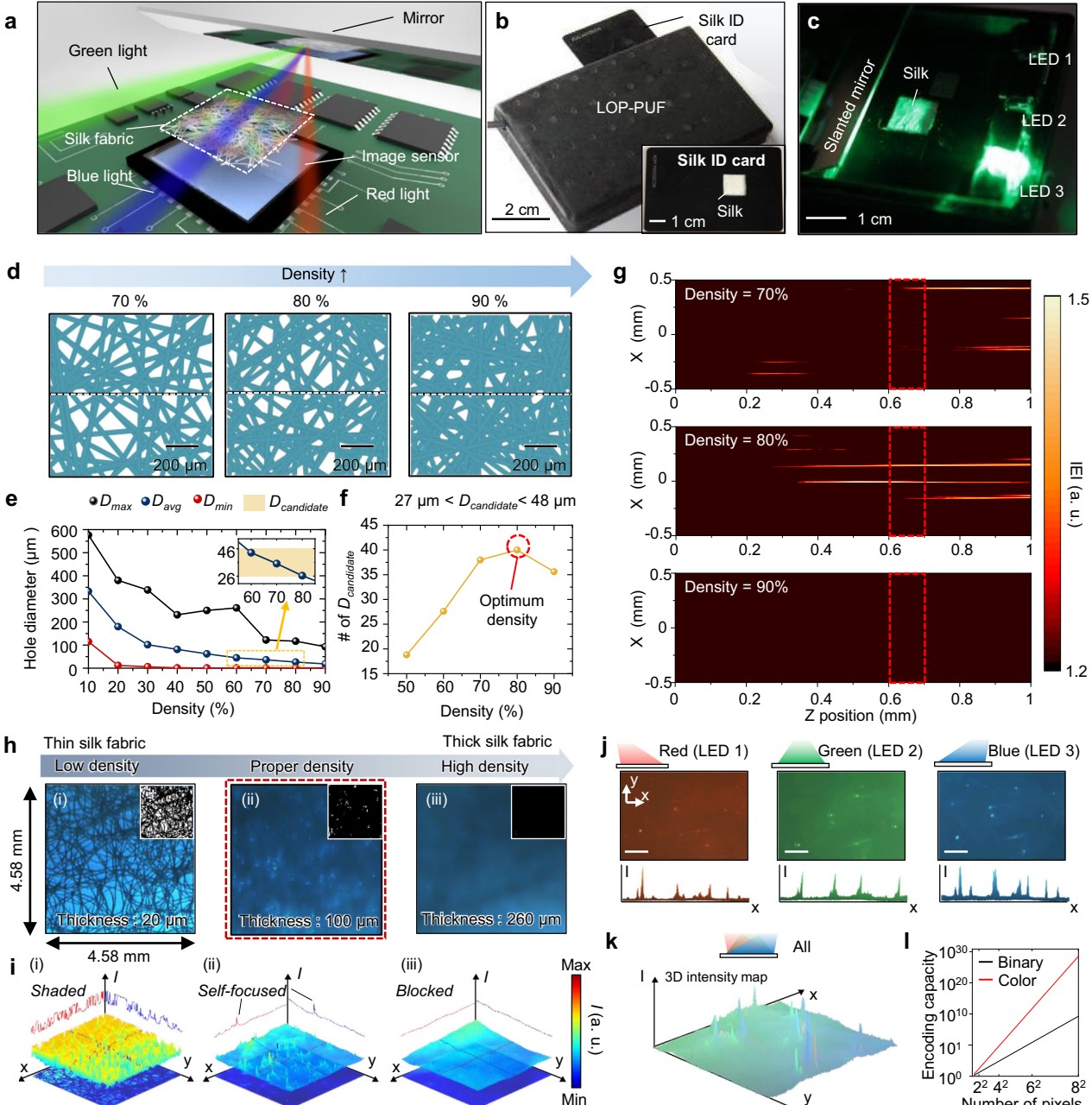

**Fig. 3 Lens-less, optical, and portable physical unclonable function (LOP-PUF). a** Schematic illustration of the proposed LOP-PUF module. **b** Optical image of the sealed LOP-PUF. The inset shows a silk identification (ID) card. **c** Photograph of the proposed LOP-PUF module in the working status. **d** Three virtual fiber media with density levels of 70, 80, and 90%. The dashed lines represent the one-dimensional simulation domain for (**g**). **e, f** Opening diameter and $D_{candidate}$ value as a function of density. **g** Electric field profiles of the three virtual fiber density levels of 70, 80, and 90%. The digitized |E| signifies the captured photo-signal by the image sensor. **h** Obtained raw data from the LOP-PUF using silk with different density levels. In the red dashed box, the self-focused spots are shown. Other areas display the absence of focal spots. The inset images are binarized raw data. **i** Three-dimensional normalized intensity for three density levels of silk cocoons. **j** Lens-less images (top) and cross-sectional intensity profiles (bottom) under illumination by three separate, colored LEDs from three different angles (−15, 0, and +15°). The scale bars are 300 μm. **k** Three-dimensional intensity map for illumination under all light sources. **l** Encoding capacity of the LOP-PUF using tri-colors (red) and mono-color (black).

during the 7 days (Fig. 4h and Supplementary Fig. 13). Moreover, the silk material shows good flexibility against mechanical and thermal stress (Supplementary Figs. 14 and 15). These results imply that the utilization of the LOP-PUF in highly humid environments requires the use of non-biodegradable fibrous medium made by water-resistant polymers; however, for indoor environments, the silk material can serve as PUF-tag for a long time.

**Performance of LOP-PUFs.** Digitized keys are extracted from 15 silk ID cards using a LOP-PUF module equipped with three LEDs of red, green, and blue light sources at different angles (i.e., −15° 0° and +15°) (Fig. 5a). The obtained bit sequences for all silk ID cards are shown in Supplementary Fig. 16. The silk ID cards are prepared by gluing native silk in a holder (Supplementary Fig. 17). Each response is extracted with several image processes from the captured raw data to unique bitmaps

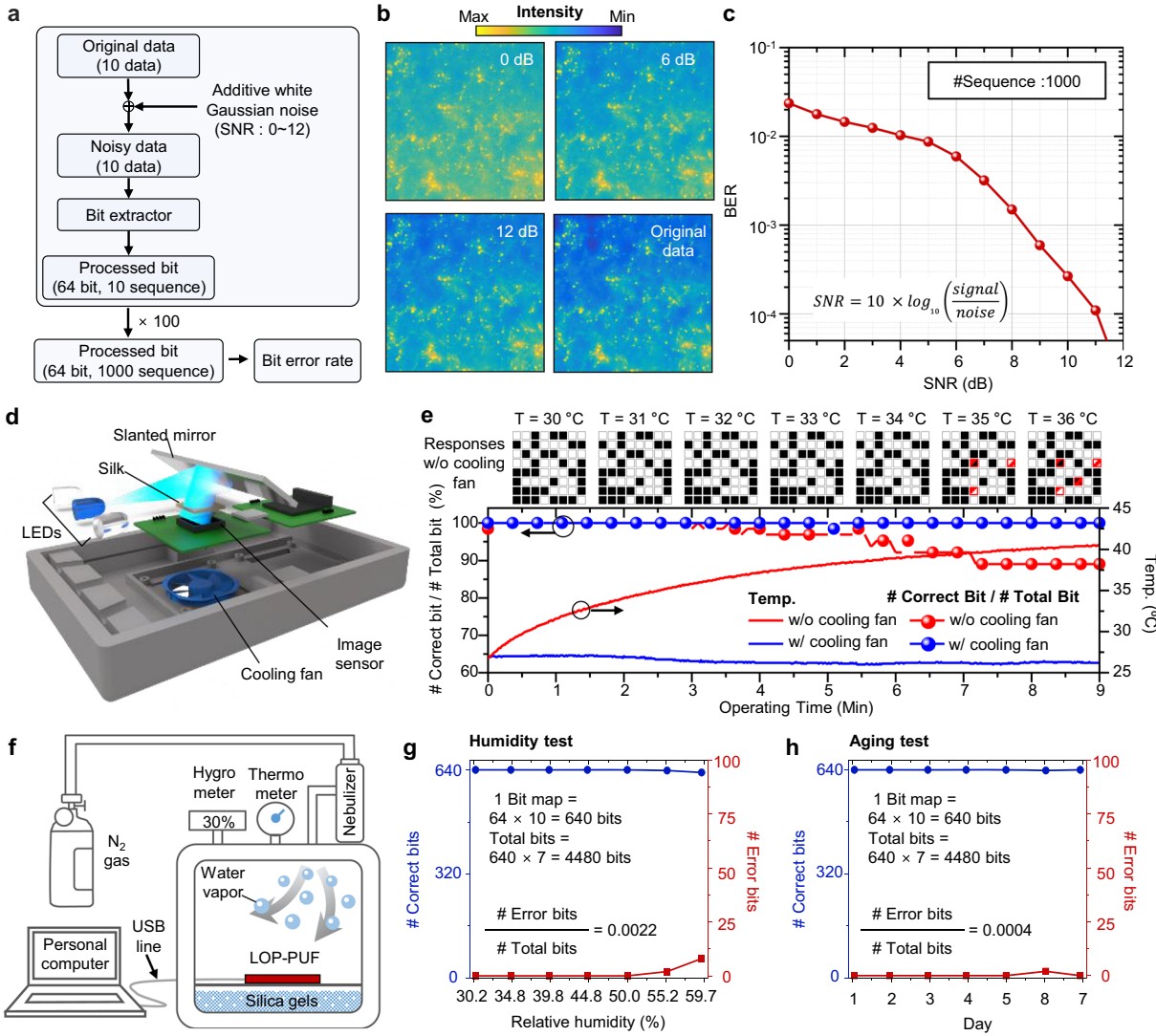

**Fig. 4 Reliability test of the LOP-PUF. a** Flow chart of signal-to-noise rate (SNR) obtained by introducing additive white Gaussian noise (AWGN) computationally. **b** Original data obtained under the blue light illumination with the center wavelength of 467 nm and treated data with AWGN to satisfy SNRs of 0, 6, and 12 dB. **c** Graph of bit error rate according to SNR. **d** Illustration of the LOP-PUF module with a cooling fan, which can reduce thermal noise. **e** (top) Obtained responses at various temperature conditions (i.e., 30, 31, 32, 33, 34, 35, and 36 °C). (bottom) Ratio of number of correct (original) bits to error bits over temperature variations as a function of the operating time of the LOP-PUF module. The red line and blue line represent the data without and with cooling fan, respectively, according to the operation time. **f** The schematic of measurement setup for humidity control. The LOP-PUF module is placed in a enclosed chamber with nebulizer, hygrometer, and silica gel. The silica gel is used to set the initial humidity. **g** Ratio of the number of correct bits to error bits depending on the relative humidity (i.e., 30.2, 34.8, 39.8, 44.8, 50.0, 55.2, and 59.7%). **h** Ratio of the number of correct bits to error bits by date over a week under room temperature and ~30% humidity.

(Supplementary Figs. 18 and 19). To evaluate our LOP-PUF system, the bit uniformity, reproducibility (intra-device HD), uniqueness (inter-device HD), 2D correlation of each PUF response, and randomness are analyzed. The bit uniformity is defined as

$$\text{Bit uniformity} = \frac{1}{s}\sum_{l=1}^{s}K_l, \qquad (1)$$

where $K_l$ is the $l$th binary bit of the key and $s$ is the key size. Each bit sequence is extracted with a von Neumann extractor, which enhances the bit uniformity up to 0.4972 (Fig. 5b).

The reproducibility of the PUF responses is evaluated by calculating the intra-device HD when the same challenge is applied to the same PUF-tag. The HD is calculated by a different number of positions when two bit streams $K_i$ and $K_j$ with a $s$-bit

key are compared.

$$\text{Hamming Distance (HD)} = \frac{\#(K_i \neq K_j)}{s}, \qquad (2)$$

$$\text{Intra-device HD} = \frac{1}{m}\sum_{t=1}^{m}\frac{\text{HD}(K_i, K_{i,t})}{s}, \qquad (3)$$

where $K_{i,t}$ represents the $s$-bit keys of the $i$th PUF device at $t$th time among $m$ different acquisition numbers. The uniqueness of the PUFs is tested by evaluating the inter-device HD, which shows the difference in the bitstreams between two different PUFs.

$$\text{Inter-device HD} = \frac{2}{q(q-1)}\sum_{i=1}^{q-1}\sum_{j=i+1}^{q}\frac{\text{HD}(K_i, K_j)}{s}, \qquad (4)$$

where $K_i$ is the binary bit of the key in the $i$th PUF device among

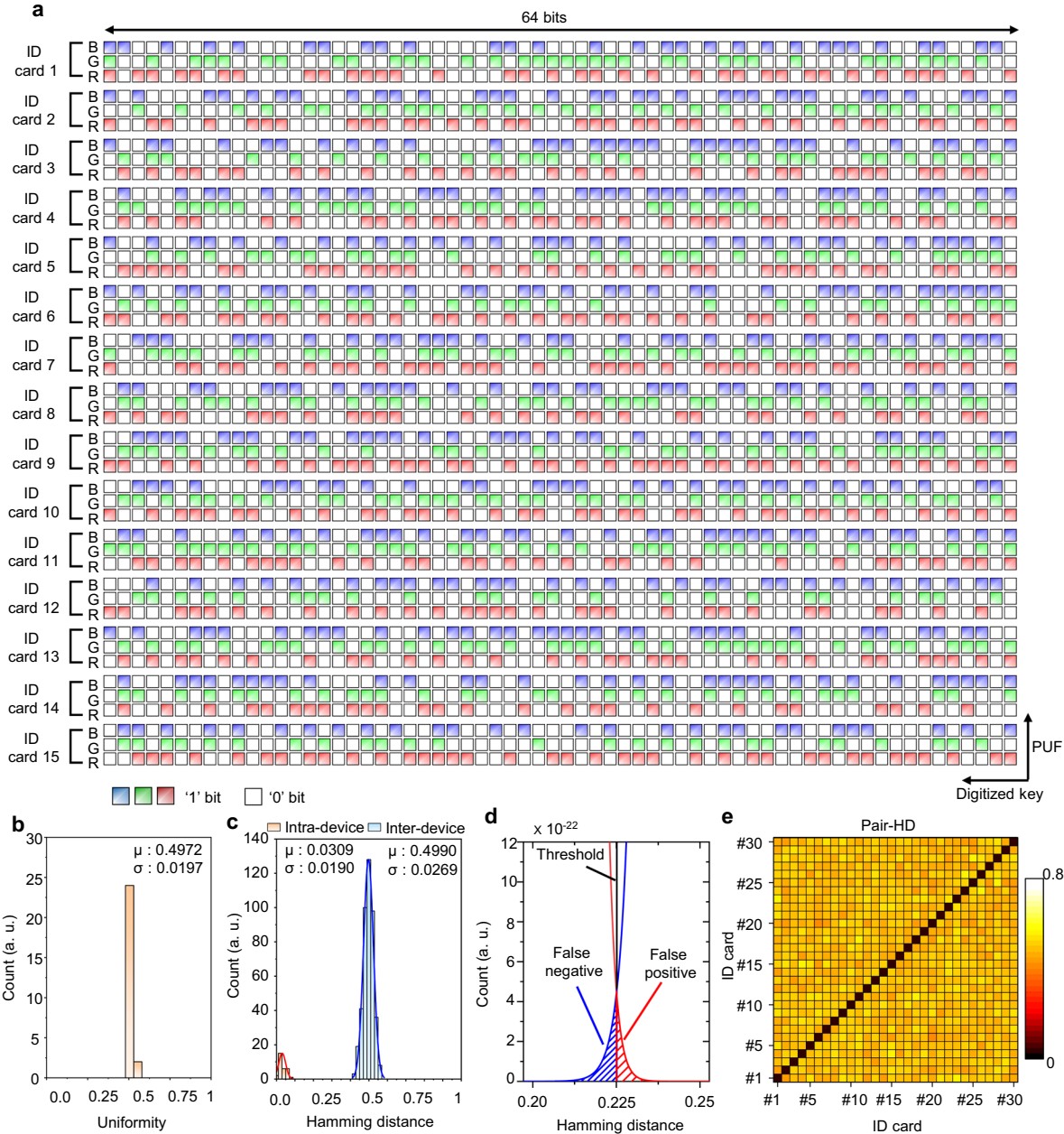

**Fig. 5 Generated bits and features of the LOP-PUF. a** Fifteen bit sequences obtained by the LOP-PUF for three spectrally separate LEDs. The three LEDs with red (R), green (G), and blue (B) light sources are positioned at an angle in the LOP-PUF to illuminate the light obliquely. A total of the obtained bit sequences (i.e., 1–30 of ID cards) are shown in Supplementary Fig. 18. **b** Bit uniformity of the response bit data with '0' and '1' obtained by counting the number of digits for each silk ID card. **c** Inter-device HD and intra-device HD of each PUF response. **d** Magnified intra-device and inter-device HD for a threshold of authentication. **e** 2D correlation of the LOP-PUF using 30 silk ID cards with nine responses. The black solid lines delineate each silk ID card and the color bars indicate the colors of the LEDs.

$q$ different PUF devices, $K_j$ is the binary bit of the key in the $j$th PUF device, and $s$ is the size of the key. In an ideal PUF system, the intra-device HD is 0 for stable acquisition and the inter-device HD is 0.5 for uniqueness. The estimated intra-device HD exhibits a relatively low mean value of 0.0309 for 30 different silk ID cards over the same 10 challenge cycles. The LOP-PUF presents high reproducibility owing to the high-intensity contrast in the silk ID card resulting from the 'self-focusing effect'. In addition, the inter-device HD is estimated from 30 silk ID cards over 9 challenges and has a mean value of 0.4990 (Fig. 5c). With the calculated intra-device HD and inter-device HD values, we estimate a threshold of the authentication mode. Gaussian fitting is used to determine the overlapped area (Fig. 5d). The false-

negative rate is the probability that an initially authenticated PUF fails authentication when the threshold is 0.225.

Moreover, the encoding capability of our LOP-PUF is estimated theoretically and practically. The conventional PUF systems can have an encoding capacity of $c^s$, where $c$ is bit states and $s$ is the size of bit sequence[36,37]. Our LOP-PUF has a $c$ value of 2 and an $s$ value of 768; hence, the theoretical encoding capacity is $2^{768}$. However, many PUF systems have estimated their encoding capacity by considering independent bit elements in bit sequences, which is called the 'degree of freedom' (DoF):

$$\mathrm{DoF} = \frac{\mu(1-\mu)}{\sigma^2}, \tag{5}$$

**Table 1 Summary of the randomness tests of binary sequences generated from the LOP-PUF using silk.**

| [a]NIST statistical test | [b]p-value | Proportion | Result |
|---|---|---|---|
| Frequency | 0.001873 | 133/135 | Pass |
| Block frequency | 0.573321 | 134/135 | Pass |
| Cumulative sums | 0.000730, 0.021542 | 133/135, 134/135 | Pass |
| Runs | 0.382509 | 134/135 | Pass |
| Longest run | 0.998338 | 135/135 | Pass |
| Serial | 0.369488, 0.320255 | 132/135, 134/135 | Pass |
| Approximate entropy | 0.573321 | 134/135 | Pass |

[a]NIST tests are performed using 135 sequences of 128 bits each such that 17,280 bits (i.e., digitized keys) collected from 30 different silk ID cards for the LOP-PUF are tested. The chi-squared ($\chi^2$) distribution is used to compare the goodness-of-fit of the p-value distribution of the blocks from the entire bitstream to the expected distribution. The bitstream is considered to be random only if the p-value ≥0.0001.
[b]If the proportion exceeds the minimum rate (>130/135) for each test, it is considered a pass.

where $\mu$ is the mean of inter-device HD and $\sigma$ is the standard deviation of inter-device HD. The LOP-PUF possesses a *DoF* of 345 ($\approx 0.4990 \times (1 - 0.4990)/0.0269^2$)[2], which shows an encoding capacity of $2^{345}$ ($\approx 7.1672 \times 10^{103}$). The pairwise HD (Pair-HD) for 30 silk ID cards is estimated over three challenges to support the uniqueness (Fig. 5e). Except for the diagonal areas in the contour, the Pair-HDs are ~0.5 on average.

To further confirm the randomness of the extracted bits, we use the NIST randomness test suite (NIST SP 800-22), which is a statistical estimation method for validating random numbers[38]. We combine all extracted bit data ($64 \times 9 \times 30$) corresponding to the responses of red, green, and blue LEDs for 30 silk ID cards (Supplementary Fig. 20). Seven statistical evaluations for randomness are performed and all of the tests are successfully passed (Table 1). The bit extraction time from random seed images is 0.422 s for each LOP-PUF. In addition, the LOP-PUF requires low power consumption: the operating time is estimated to be ~27 h when using a 7500 mAh portable battery (Supplementary Fig. 21).

**Authentication and encoding capability of the LOP-PUF.** The proposed LOP-PUF has the capability of a 'user authentication' mode (Fig. 6a). In this mode, challenges are exploited to check the identification of users by taking the responses from the data center; if each challenge–response pair (CRP) was validated, the identification succeeded. Since the encoding capacity was incredibly high in the LOP-PUF, the used CRP could be discarded. One of the simple attack methods, which is called Brute Force attack, uses random substitution with a number of cases according to the key size (Fig. 6b). For reliable statistical analyses, break times ($B_{time}$), which are the time required for successful access using a fake key in a single attempt, were averaged over ten attempts. The security level for authentication application was measured against a simple attack in our LOP-PUF using silk, showing an average $B_{time}$ of $5 \times 10^{41}$ years (Fig. 6c). This result indicated that the LOP-PUF using three-color space (i.e., three channels) was an unbreakable security system since the time required for fake authentication was an unrecognizable time for human beings.

Another mode of the LOP-PUF is a 'software and hardware binding' (SHB) mode (Fig. 6d). For example, image data are encoded and decoded using the extracted key by the LOP-PUF with silk. In the SHB mode, the original data are encrypted and stored with the extracted key. In addition, to present the output data, the extracted key should be used; however, if the encoded data are displayed without the PUF, the output data produce a

random noise image. Figure 6e exhibits five cases of encoding/decoding for the 'Skein image' using five extracted keys (i.e., R1, R2, R3, R4, and R5) with different uniformity levels (i.e., 10, 30, 50, 70, and 90% of '1'-bit ratios). Except for the 50% ratio (i.e., R3), all other cases are artificially generated to compare the PUF encoding performance. R3 is the extracted data from our LOP-PUF using silk. See the "Methods" section for the process of artificial uniformity key generation in detail.

When encoding with R3, the original data (OD) are encoded such that it is impossible to recognize the 'skein image' because the uniformity of R3 is 50%. In the case where the uniformity is biased, the encoded data (ED1, ED2, ED4, and ED5) exhibit that the feature of the 'skein image' remains, which is vulnerable to potential cyber-attacks. However, the decoded data (DD) are completely restored because the encoding and decoding processes use the same key in each case. The HDs of OD-ED and OD-DD are estimated quantitatively (Fig. 6f). The case of the silk ID card, which has a uniformity of 0.5, indicates the best encoding performance because the HD of OD-ED is not biased to '0' or '1' bits. The detailed encoding/decoding process is presented in the "Methods" section and Supplementary Fig. 22.

**Conclusions.** In conclusion, we present a compact, small size, and low-cost LOP-PUF system with fibrous medium ID cards. A stochastic distribution of microfibers in native silk leads to a 'self-focusing effect' under an incoherent light source (i.e., an LED). In addition, the nanofibrillar structures in each micro-fiber significantly improves the light intensity contrast between the background and focal spots owing to the strong scattering. Our theoretical analyses systematically explore these optical phenomena using numerical experiments of single fibers and fiber bundles. These novel optical features could easily implement the module of a lens-free optical PUF by placing a silk ID card on the image sensor. Moreover, our module increases the amount of CRPs by using spectral multiplexing. Thus, the proposed LOP-PUF satisfies the key characteristics of a PUF in terms of bit uniformity, reproducibility, uniqueness, and randomness. Finally, we demonstrate the applicability of the LOP-PUF system in two modes: authentication and data encryption.

## Methods

**Implementation of the LOP-PUF module.** A commercial 3D printer (Duplicator 8, Wanhao, China) was used to produce the LOP-PUF module. Black resin was selected as the base material to block ambient light from the outside module. A commercial image sensor with an FPGA chip (MT9J003, On Semiconductor Corp., USA) captured and transferred the obtained images to a personal computer in real time. The image sensor recognized color information because the Bayer color filters were composed of one red (R), two green (G), and one blue (B) pixels. The communication between the image sensor and the personal computer was conducted through a USB cable, and an open-source programming language (Python 3.8.5) was used to capture the images. Three tricolor LEDs (LEDRGBE 627.5/525/467.5 nm, Thorlabs, Inc., Germany) were racked on the 3D-printed module to illuminate the separate wavelengths at different incident angles. The fabricated silk ID card was inserted into the module to directly contact the image sensor. Native silk cocoons produced by wild-type silkworms (*Bombyx mori*) were used.

**Random bit extraction.** MATLAB (Mathworks, Inc., USA) was used for image processing and bit extraction. The captured images had a RGB-color space. For image processing, the image was decomposed using three channel images. Next, each image was exploited to generate bitmaps. To equalize the LED illumination, the original image and blurred image through the Gaussian filter were subtracted from each other. Moreover, a binning process was performed to reduce the peak/edge noise. The detailed binning method was shown in Supplementary Fig. 23. An image size of 2048 pixels × 2048 pixels was binned to an image size of 32 pixels × 32 pixels. A threshold was applied to the binned image for digitization of the data. Finally, von Neumann debiasing was conducted in each bit column stream (i.e., 32 × 1 bits). The detailed von Neumann debiasing is shown in Supplementary Fig. 24. If the debiased column bits were more than four bits, the first four bits were extracted. Otherwise, the processing moved to the

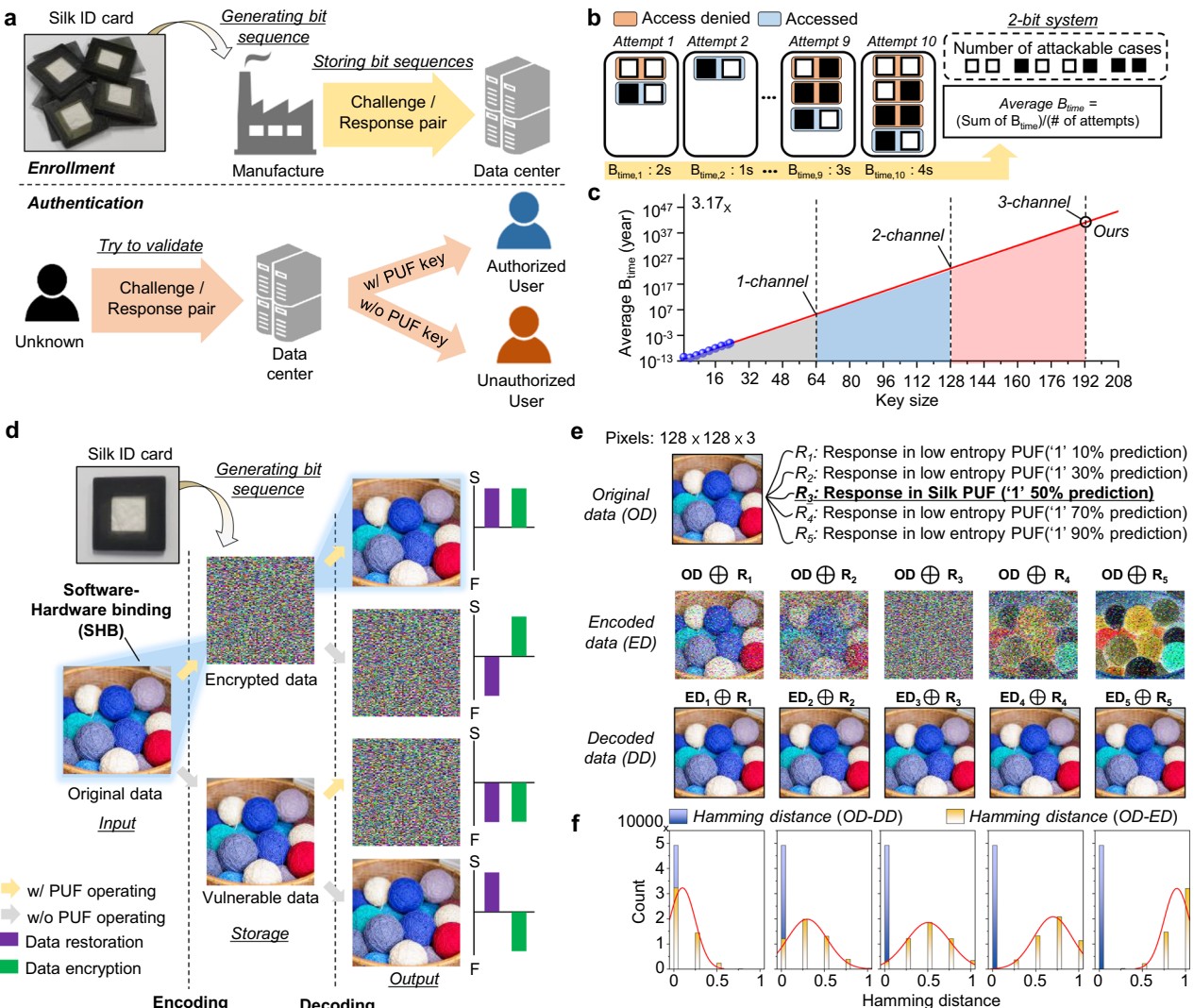

**Fig. 6 Applications of the LOP-PUF: authentication and data encoding/decoding. a** Schematic illustration for authentication. 'Enrollment' is the step where the manufacturer registers the PUF keys from silk ID cards in the data center. 'Authentication' is the process for the validation of unknown users using stored PUF keys. **b** Illustration of a brute force attack test for estimating the defense ability of the authentication system by substituting a random key for fake authentication. The average break time, $B_{time}$, is calculated using the individual $B_{time}$ values for ten attempts. **c** Graph of the average break time using the brute force attack in our proposed system as a function of the key size. Using three-color space allows the creation of a large number of key sizes for a strong security level. **d** Description for an application of data encryption in the LOP-PUF. Successful data encoding and decoding with encryption are possible only when software and hardware binding is ensured. With operating PUF, input data are stored as encrypted and the stored data are decoded properly. Without operating PUF, the input data are vulnerable and the stored data are illegible. The purple and green boxes indicate success or failure of the data restoration and encryption depending on the operation of the PUF. The letters 'S' and 'F' mean success and fail, respectively. **e** Data encryption and decryption in the pictured materials using five responses with different '1'-bit ratios of 10, 30, 50, 70, and 90% where the PUF with a half ratio of '1' bit is generated by the LOP-PUF using silk. **f** Hamming distance (HD) of the original data vs. encoded data (OD-ED) and the original data vs. decoded data (OD-DD). The red solid lines refer to the HD distributions of the OD-ED.

adjacent bit column sequence and extracted four bits. This process was repeated until 64 bits were collected.

**Encoding decoding key generation**. To generate a large bit size, we combined all extracted bitmaps from 30 silk ID cards. The three channel responses consisted of $64 \times 9 \times 3$ corresponding to the bit size, the number of samples, and the number of light angles, respectively. The combined bits were repeated twice horizontally and ten times vertically to fit the size of the target data. The processed bitmap was cropped to a size of $128 \times 256$, and then it was repeated twice horizontally and vertically. Finally, the size of the $256 \times 512$-bit data was reshaped to $128 \times 128 \times 8$ to apply the XOR operation with the target data.

**Artificial uniformity key generation**. To generate an artificial random key, the MATLAB function 'rand' was used, which yielded a normal random number in the range of 0 to 1. By applying a threshold, we extracted a digitized key with a size of

$64 \times 27$, which was the same key size as the key form of the silk ID card. For specific uniformity, various thresholds were applied as 0.1, 0.3, 0.7, and 0.9. In addition, a conditional statement was used to check the bit uniformity.

**Data encoding and decoding process**. The data encoding and decoding process was performed using MATLAB (Mathworks, Inc., USA). For encoding, the image was decomposed into three channel images (red, green, and blue). Each channel image consisted of 128 pixels $\times$ 128 pixels with an 8-bit level (the image size was $128 \times 128 \times 8$). For encoding and decoding, each channel image was subjected to the XOR operation with processed bitmaps. The detailed encoding method is shown in Supplementary Fig. 22.

**Random nanohole and random fiber bundle generation**. The random simulation domain was defined as a blank matrix in the initial state. One element of the matrix was a unit dimension. The virtual rectangular fibers with nanoholes

were defined to fill the domain with an arbitrary center position and angle. The created fibers with nanoholes were recorded as '1' in the matrix. To check the density of the medium, a ratio of 1 to 0 was calculated for every generation. The detailed simulation method is shown in Supplementary Figs. 25 and 26.

**Reporting summary**. Further information on research design is available in the Nature Research Reporting Summary linked to this article.

## Data availability

The image data are available in a GitHub repository at https://github.com/seok9643/Silk_PUF. The extra data and code supporting the findings of this study are available from the corresponding authors upon reasonable request. Source data are provided with this paper.

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

## Acknowledgements

This research was supported by the National Research Foundation of Korea (NRF-2020R1A2C2004983/2018M3D1A1058997) and by the GIST Research Institute (GRI) grant funded by the GIST in 2021. This work was also supported by the Institute of Information & Communications Technology Planning & Evaluation (IITP) through a grant funded by the Korean government (MSIT) (No. 2020-0-01000, Light field and LiDAR sensor fusion systems for full self-driving) and by the United States Air Force Office of Scientific Research (FA2386-17-1-4072).

## Author contributions

M.S.K. and G.J.L. developed the idea of this study. M.S.K. and G.J.L. worked on the device fabrication, the optical measurements, and the key generation. J.W.L., S.H.C., Y.L.K. and Y.M.S. participated in the analyses. M.S.K. and G.J.L. wrote the paper. Y.L.K. and Y.M.S. directed the overall research. All authors contributed to writing the manuscript.

## Competing interests

The authors declare no competing interests.
