## [Peer Review File · Nature Communications]

Revisiting silk: a lens-free optical physical unclonable functionREVIEWER COMMENTS

Reviewer #1 (Remarks to the Author):

This manuscript demonstrates a PUF system using native silk based on its inherent randomness of microfiber distribution. The authors used the “self-focusing effect” of silk to construct a compact, low-cost and lens-free LOP-PUF system, which is a novel and interesting idea. The removal of objective lens and complicated optical paths is significant for the practicality of optical PUFs. The authors also verified the reproducibility and uniqueness of LOP-PUF, and demonstrated two application modes of the system. The proposed PUF is equipped with multiple challenge-response pairs by using spectral multiplexing. It is also interesting to establish a relationship between Brute Force attack for the PUF and corresponding break times. In short, this work is of great interest and shows a good data abundance. Therefore, I recommend the acceptance of this work for publication on Nature Communications after addressing some major concerns shown below.

1. The encoding capacity is an important parameter for PUF systems. The authors did not clarify the encoding capacity of the LOP-PUF system.
2. The authors claimed that the silk exhibits superior mechanical properties. Can the authors show the mechanical stability of PUF for bit-exact readout?
3. The silk is moisture sensitive and easy to biodegrade, and this may affect the stability and aging of the PUF. The authors should investigate their effects in this work.
4. The signal-to-noise ratio for the PUF readout should be mentioned.
5. To reconcile a noisy response, the unreliability (that is, the bit error rate) of PUFs is a major hurdle for applications and should be shown.
6. As far as I can see, the focal spots distribute sparsely in the obtained images (e.g., Figure 3I, Figure S6). This means there will be many “0” in digital keys, which will significantly reduce the encoding capacity of LOP-PUF. The authors should make a discussion on this issue. In the caption of Figure 3I, “four densities of silk fabric” should be “three densities”?
7. Figure 1D, the picture of $dIS=0$ seems to exhibit focal spots at the same sites as $dIS=0.5$ mm, despite of a weaker intensity. While the upper circle in $dIS=1.0$ mm does not show a focal spot for me. Can the authors explain it?
8. In the reproducibility and uniqueness verification experiments, the sample volume of 16 silk ID cards is not sufficient to get reliable results. The authors should enlarge their sample capacity.
9. The authors used HD to represent the reproducibility and uniqueness of LOP-PUF, but detailed description on HD calculation is absent.

10. The authors did not give detailed descriptions on how to get the threshold for PUF digitization.
11. In Method section, some SI Figure number is wrong. For example, line 315, "Fig. S13" should actually be Fig. S14. Please check the whole manuscript to ensure consistency.

Reviewer #2 (Remarks to the Author):

This paper develops a lens-free, optical, and portable PUF based on stochastically manifested diffraction using native silk fibers. By optimizing the distance between a silk PUF-tag and an image sensor, this simple apparatus engineering easily formed random light-spot patterns with high-intensity contrast. Overall, this work is interesting and novel. Most of the paragraphs are clearly written. Here I only have a major concern about the PUF performance. For example, the temperature dependent CRPs and the power consumption are missing. Please refer to my detail suggestions:

1. A high quality PUF has also to be reliable. This means that at any point in time, under any operation conditions, the CRPs have to be unchanged. Although the authors have estimated the intra-device HD to be 0.1282, the temperature may induce changes of the width of the stochastic randomly distributed holes produced by silk fibers, which may further result in changes of the generated bitstreams. In their system, how large is the sensitivity to temperature.
2. This work also lacks important information such as power consumption, speed and operating voltage of implemented silk fiber-based optical PUF.
3. It would be better to cite latest works as introducing a certain technology. For example, when authors mentioned various mechanism-based PUFs, magnetics PUF, very recently, was reported using the methods of spin-orbit-torque driven magnetic switching (Journal of Applied Physics 128, 033904 (2020), IEEE Electron Device Letters 42, 597 (2021)...) or interface anisotropy modulation (18, 7211(2018))

Reviewer #1:

Summary Comments: *This manuscript demonstrates a PUF system using native silk based on its inherent randomness of microfiber distribution. The authors used the “self-focusing effect” of silk to construct a compact, low-cost and lens-free LOP-PUF system, which is a novel and interesting idea. The removal of objective lens and complicated optical paths is significant for the practicality of optical PUFs. The authors also verified the reproducibility and uniqueness of LOP-PUF, and demonstrated two application modes of the system. The proposed PUF is equipped with multiple challenge-response pairs by using spectral multiplexing. It is also interesting to establish a relationship between Brute Force attack for the PUF and corresponding break times. In short, this work is of great interest and shows a good data abundance. Therefore, I recommend the acceptance of this work for publication on Nature Communications after addressing some major concerns shown below.*

Our response to summary comments: We sincerely appreciate your valuable comments, which were helpful in improving the quality of our manuscript significantly. We have revised our manuscript accordingly and our responses are described below.

Comment #1: *The encoding capacity is an important parameter for PUF systems. The authors did not clarify the encoding capacity of the LOP-PUF system.*

Our response to Comment #1: We appreciate the reviewer for the fruitful comment that improves the quality of the manuscript. Ideally, PUF systems have an encoding capacity of c^s , where c represents bit states and s is the bit sequence size. Our system has a c value of 2 and an s value of 768; hence, the theoretical encoding capacity is 2^{768} . However, many PUF systems have estimated their encoding capacity by considering independent bit elements in bit sequences, called as degree of freedom (DoF):

$$DoF = \frac{\mu(1 - \mu)}{\sigma^2}, \quad (1)$$

where μ and σ are a mean and a standard deviation of inter-device hamming distance (inter-device HD) for 30 different LOP-PUFs, respectively [R1, R2]. In our revised manuscript, we have increased the number of silk ID cards to further raise the reliability of PUFs as suggested in *Comment #8*. The Gaussian distribution of the updated inter-device HD values shows a mean value of 0.4990 and a standard deviation of 0.0269, as shown in **Figure R1**. Based on these values, we obtained a DoF of 345 ($\approx 0.4990 \times (1 - 0.4990) / 0.0269^2$). The actual encoding capacity is defined as c^{DoF} ; therefore, our system shows an encoding capacity of 2^{345} ($\approx 7.1672 \times 10^{103}$).

Figure R1. Histogram of inter-device Hamming Distance (HD) for 30 different LOP-PUFs. Inter-device HD characterizes the device uniqueness of LOP-PUFs. For inter-device HD, a Gaussian fit of the histogram returns $\mu = 0.4990$ and a standard deviation (SD; σ) of 0.0269 where the probability density is of Gaussian distribution.

We also compared the encoding capacity of our system with that of other PUF systems reported in previous works (see **Table R1**). The optical PUFs have a higher encoding capacity than other types of PUFs [R3]. Additionally, the encoding capacity of conventional optical PUFs depends on the number of pixels. To increase the encoding capacity of optical-PUF, a sophisticated imaging setup is necessary to capture more information. Although LOP-PUF uses no additional imaging system, the encoding capacity shows a relatively high value compared to the recently reported biomaterial-based optical PUF using the imaging system [R4]. Based on the above, we have clarified the encoding capacity of our PUF system in the revised manuscript.

Table R1. Encoding capacity comparison of our LOP-PUF system with other PUF systems.

Type of PUF	Material	Readout system	Encoding capacity	Ref.
Optical-PUF	Inorganic (polymer)	Microscope lens	$\approx 10^{225}$	[R5]
Optical-PUF	Inorganic (TiO ₂)	Macro lens	$\approx 2.5 \times 10^{120}$	[R6]
Optical-PUF	Organic (silk)	Zoom lens	$\approx 1.3292 \times 10^{36}$	[R4]
Optical-PUF	Inorganic (MoS ₂)	Microscope lens	$\approx 3.5490 \times 10^{131}$	[R7]
Optical-PUF	Organic (silk)	Lens-free	$\approx 7.1672 \times 10^{103}$	Ours

References

- [R1] Pappu, Ravikanth, et al. "Physical one-way functions." *Science* 297.5589 (2002): 2026-2030.
- [R2] Hu, Zhaoying, et al. "Physically unclonable cryptographic primitives using self-assembled carbon nanotubes." *Nature nanotechnology* 11.6 (2016): 559-565.
- [R3] Wali, Akshay, et al. "Biological physically unclonable function." *communications Physics* 2.1 (2019): 1-10.
- [R4] Leem, Jung Woo, et al. "Edible unclonable functions." *Nature communications* 11.1 (2020): 1-11.
- [R5] Wu, Bai-Heng, et al. "Grain boundaries of self-assembled porous polymer films for unclonable anti-counterfeiting." *ACS Applied Polymer Materials* 1.1 (2018): 47-53.
- [R6] Arppe-Tabbara, Riikka, Mohammad Tabbara, and Thomas Just Sørensen. "Versatile and validated optical authentication system based on physical unclonable functions." *ACS applied materials & interfaces* 11.6 (2019): 6475-6482.
- [R7] Park, Jaeseo, et al. "Disordered heteronanostructures of MoS₂ and TiO₂ for unclonable cryptographic primitives." *ACS Applied Nano Materials* 4.2 (2021): 2076-2085.

Our revision to the manuscript:

(Line 12, Page 14: in the revised main text)

“Moreover, the encoding capability of the LOP-PUF was estimated theoretically and practically. The encoding capacity of conventional PUF systems is defined as c^s , where c is bit states and s is the size of the bit sequence. In the LOP-PUF, c and s correspond to 2 and 768, respectively, resulting in the theoretical encoding capacity of 2^{768} . However, many PUF systems have estimated their encoding capacity by considering independent bit elements in bit sequences, called as the ‘degree of freedom’ (DoF):

$$DoF = \frac{\mu(1-\mu)}{\sigma^2}, \quad (5)$$

where μ and σ are a mean and a standard deviation of inter-device hamming distance (inter-device HD) for 30 different LOP-PUFs, respectively. The DoF of the LOP-PUF is 345 ($\approx 0.4990 \times (1 - 0.4990) / 0.0269^2$), and thus its mutual encoding capacity is 2^{345} ($\approx 7.1672 \times 10^{103}$).”

Comment #2: *The authors claimed that the silk exhibits superior mechanical properties. Can the authors show the mechanical stability of PUF for bit-exact readout?*

Our response to Comment #2: We appreciate the reviewer for the fruitful comment that improves the quality of the manuscript. In our PUF system, the important factor for bit generation is micro fiber distribution; therefore, we analyzed the fiber distribution for mechanical stability using bending cycle tests. In the experiment, we analyzed the silk after every iteration of the bending test and finally compared the first image (*i.e.*, before bending)

with the other images (*i.e.*, after bending) in terms of the 2-dimension correlation coefficient (**Figure R2a**). To confirm the observation area of silk, we made observation keys using a commercial laser cutter (Left; **Figure R2b**). The bending test was performed using an aluminum rod with a radius of 2.5 mm (Right; **Figure R2b**). **Figure R2c** shows the raw image of captured silk with an objective lens with a magnification of 10. The red dashed box indicates the observation area.

We captured four images: before bending, after 1st bending, 10th bending, and 100th bending. Although we made the alignment keys to consistently see the same position in the silk, it is difficult to capture exactly the same images due to repeated loading/unloading of silk on the microscope stage. To address this issue, we aligned the four images using the ‘Auto-align’ function of a commercial software (Adobe Photoshop CS6). **Figure R2d** shows the aligned four images, which are almost identical images. We then mathematically compared the pristine image with the images after bending using correlation coefficients (**Figure R2e**). The result demonstrates the durability of the silk and confirms that even bending 100 times barely influences the micro fiber distribution. After 100 times bending, the high correlation coefficient value of ~ 0.8932 between the two images supports that silk is a suitable material for our PUF system.

To offer further information to the readers, we have added these experimental results in the Supplementary Information.

Figure R2. (a) Schematic illustration of measurement setup for bending test. (b) Photographs of observed silk with alignment keys (left) and bent status of silk with the radius of curvature of 2.5 mm (right). (c) Silk observed using 10 \times objective lens. Red dashed box indicates the observation area. (d) Raw images with different bending cycles (*i.e.*, before bending, 1st, 10th, and 100th bending). (e) Correlation coefficients of raw images with bending compared with the raw image before bending.

Our revision to the manuscript:

(Line 5, Page 11: in the revised main text)

“Moreover, the native silk shows good flexibility against mechanical and thermal stress (Supplementary Figs. 14 and 15)”

(Supplementary Figure 14: in the revised Supplementary Information)

Supplementary Figure 14. (a) Schematic illustration of measurement setup for bending test. (b) Photographs of observed silk with alignment keys (left) and bent status of silk with the radius of curvature of 2.5 mm (right). (c) Silk observed using 10× objective lens. Red dashed box indicates the observation area. (d) Raw images with different bending cycles (*i.e.*, before bending, 1st, 10th, and 100th bending). (e) Correlation coefficients of raw images with bending compared with the raw image before bending.

Comment #3: The silk is moisture sensitive and easy to biodegrade, and this may affect the stability and aging of the PUF. The authors should investigate their effects in this work.

Our response to Comment #3: We appreciate the reviewer for the fruitful comment that improves the quality of the manuscript. As suggested, we investigated the moisture sensitivity and biodegradability of the LOP-PUF using a customized chamber with humidifier to control relative humidity (RH) (**Figure R3a**). First, we lowered the initial RH using silica gels. **Figure R3b** shows a photograph of the measurement setup, which consists of a gas cylinder, humidifier, and encapsulated chamber. The LOP-PUF system was placed in the chamber and operated under controlled RH conditions.

Figure R3c presents photographs of operating LOP-PUF module at various humidity conditions within room temperature (*i.e.*, RH = 30.2, 34.8, 39.8, 44.8, 50.0, 55.2, and 59.7 %) under the blue light illumination with the center wavelength of 467 nm. **Figure R3d** shows each extracted bit sequence for all RH. The bit extraction was conducted 10 times to confirm the stability of bit sequences. The extracted bits at RH of 30.2, 34.8, 39.8, and 44.8 % show robust results with the same bit sequences for 10 repeat measurements. Otherwise, the

extracted bits at RH of 50.0 and 59.7 % exhibit a small error rate of 0.0022 (# Error bits / # Total bits = 10/(640×7)) during 10 repeat measurements.

Figure R3. (a, b) Schematic (a) and photographs (b) of measurement setup for various relative humidity conditions. Silica gels were used to lower the initial humidity, and the

humidifier raised the humidity. (c) The obtained responses at each humidity under the blue light illumination with the center wavelength of 467 nm.

To investigate the biodegradability of silk material, we conducted measurements over a period of one week using the chamber setup under the conditions of room temperature and humidity with the blue light illumination with the center wavelength of 467 nm. In this result, an error of 1 bit appeared on the sixth day during the seven-day experiment (**Figure R4**). This result demonstrates that the silk material is a suitable material for the PUF-tag in indoor environments. For the 10 repeated measurements at each environment and over multiple days, the error rate is only 0.0004 ($\# \text{ Error bits} / \# \text{ Total bits} = 2/(640 \times 7)$), verifying the robustness of LOP-PUFs.

Figure R4. Raw data (left) and bit response (right) obtained from the LOP-PUF over seven days. The bit responses show ‘1’ bit error on the sixth day in the measurement conducted over a seven-day period. Data acquisition was repeated 10 times per measurement under the blue light illumination with the center wavelength of 467 nm.

Besides the sensitivity and biodegradability, we conducted a reliability test of LOP-PUF considering noisy response (*i.e.*, thermal noise) and bit error rate (BER), as requested in *Comments #4* and *#5*. Based on these results, we have added a new Figure 4 in the revised manuscript along with a description of the results.

Our revision to the manuscript:

(Line 15, Page 9: in the revised main text)

“*Reliability of LOP-PUFs*”

A high-quality PUF should have high reliability against external and internal factors such as thermal noise and aging issues. First, the robustness of the LOP-PUF to the thermal noise was estimated by simulating bit error rate (BER) depending on the signal-to-noise ratio (SNR). Fig. 4A illustrates the bit acquisition process to simulate the BER of the LOP-PUF. Ten-bit sequences with 64-bit sizes are obtained in one acquisition process, and this process was repeated a hundred times to acquire a thousand bit sequences because the BER test demands a number of bit sequences. Fig. 4B presents the original data as well as the data treated by white Gaussian noise with different intensities to satisfy the SNRs of 0, 6, and 12 dB. Other SNRs of treated data (2, 4, 8, 10 dB) are shown in Supplementary Fig. 9. Fig. 4C shows the BER of the LOP-PUF versus SNRs. The LOP-PUF shows a low BER ($<10^{-4}$) over the SNR of 10 dB, *i.e.*, when the intensity of signal is 10 times higher than that of noise.

To investigate the actual influence of thermal noise in the LOP-PUF, continuous operations of the LOP-PUF were conducted by measuring the temperature of the module and capturing the responses without and with temperature control (Figs. 4D and 4E). The obtained responses without and with temperature control are displayed in Supplementary Figs. 10 and 11, respectively. Without the temperature control, the image sensor of LOP-PUF heated up to ~ 35 °C in 6 min; however, with a cooling fan, the temperature maintained at ~ 27 °C throughout (Fig. 4D). The LOP-PUF without a cooling fan tends to cause error bits during the operation (Fig. 4E). This result demonstrates that the LOP-PUF without the cooling fan shows a decrease in correction rate (*i.e.*, # Correct bits / # Total bits) to 90% as the operating time increases (*i.e.*, increase in temperature of image sensor). In contrast, the cooling fan ensures that the LOP-PUF shows a stable correction rate of $\sim 100\%$.

In addition, because the LOP-PUF adopts silk material as the PUF-tag, biodegradability is an important issue as it may degrade the reliability of the LOP-PUF. A customized measurement setup was used to explore the biodegradability effect of silk (Fig. 4F). Because silk material is susceptible to humidity, our setup could control the humidity using silica gel and humidifier. The initial relative humidity (RH) was controlled at a low value, *i.e.*, 30.2%, and then it is increased up to 59.7% using the humidifier. While controlling the RH, the LOP-PUF captured the responses and the obtained responses were transformed to bit sequences. The bit extraction was conducted 10 times to confirm the stability of bit sequences.

The extracted bits at the RH of 30.2, 34.8, 39.8, 44.8, and 50.0% show robust results (*i.e.*, zero-bit error). However, the extracted bits at the RH of 55.2 and 59.7 % exhibit 2- and 8-bit errors, respectively, during 10 times generation (Fig. 4G and Supplementary Fig. 12). During the humidity test, the LOP-PUF exhibits an error rate (*i.e.*, # Error bits / # Total bits) of 0.0022. Additionally, a long-term measurement was performed for a week under the conditions of room temperature and humidity. In this result, an error of 1 bit, which corresponds to an error rate of 0.0004, appeared at the sixth day during the seven days (Fig. 4H and Supplementary Fig. 13). Moreover, the native silks show good flexibility against mechanical and thermal stress (Supplementary Figs. 14 and 15). These results imply that the utilization of the LOP-PUF in highly humid environments requires the use of non-biodegradable fibrous medium made by water-resistant polymers; however, for indoor environments, the silk material can serve as PUF-tag for a long time.

”

(Figure 4: in the revised main text)

Figure 4. Reliability test of the LOP-PUF. (A) (A) Flow chart of signal-to-noise rate (SNR) obtained by introducing additive white Gaussian noise (AWGN) computationally. (B) Original data obtained under the blue light illumination with the center wavelength of 467 nm and treated data with AWGN to satisfy SNRs of 0, 6, and 12 dB. (C) Bit error rate according to SNR. (D) Schematic of the LOP-PUF module with a cooling fan, which can reduce thermal noise. (E) (top) Bit responses of a LOP-PUF without a cooling fan at various temperature conditions (*i.e.*, 30, 31, 32, 33, 34, 35, and 36 °C). (bottom) Ratio of number of corrected and error bits and temperature variation versus operating time of the LOP-PUF module with and without a cooling fan. The red line and blue line represent the data without and with a cooling fan, respectively, according to operation time. (F) Schematic of measurement setup for humidity control. The LOP-PUF module is placed in the enclosed chamber with nebulizer, hygrometer, and silica gel. The silica gel is used to set the initial humidity. (G) Number of correct bits and error bits obtained from the LOP-PUF at different relative humidity conditions (*i.e.*, 30.2, 34.8, 39.8, 44.8, 50.0, 55.2, and 59.7 %). (H) Number of correct and error bits obtained from the LOP-PUF over seven days under room temperature and ~30% relative humidity.

(Supplementary Figure 9: in the revised Supplementary Information)

Supplementary Figure 9. The obtained raw data with the LOP-PUF system and images with artificially added noise with different SNRs (*i.e.*, 0, 2, 4, 6, 8, 10, and 12 dB).

(Supplementary Figure 10: in the revised Supplementary Information)

Supplementary Figure 10. (a) Temperature variation of an image sensor as a function of the operating time of a LOP-PUF module with and without under the blue light illumination with the center wavelength of 467 nm (optical intensity of $64 \mu\text{W}/\text{cm}^2$). (b) Maximum noise intensity (black solid line) and number of thermal noise (red solid line) as a function of the operating time of a LOP-PUF module in dark. (c) Bit maps extracted from a LOP-PUF module at various temperatures of 25-36 °C.

(Supplementary Figure 11: in the revised Supplementary Information)

Supplementary Figure 11. Bit sequences obtained by a LOP-PUF with the temperature control (*i.e.*, 27 °C).

(Supplementary Figure 12: in the revised Supplementary Information)

Supplementary Figure 12. (a, b) Schematic (a) and photographs (b) of measurement setup for various relative humidity conditions. Silica gels were used to lower the initial humidity, and the humidifier raised the humidity. (c) The obtained responses at each relative humidity value under the blue light illumination with the center wavelength of 467 nm (*i.e.*, RH = 30.2, 34.8, 39.8, 44.8, 50.0, 55.2, and 59.7 %).

(Supplementary Figure 13: in the revised Supplementary Information)

Supplementary Figure 13. Raw data (left) and bit response (right) obtained with the LOP-PUF over seven days. The bit responses show ‘1’ bit error on the sixth day in the measurement conducted over a seven-day period. Data acquisition was repeated 10 times per measurement under the blue light illumination with the center wavelength of 467 nm.

(Supplementary Figure 14: in the revised Supplementary Information)

Supplementary Figure 14. (a) Schematic illustration of measurement setup for bending test. (b) Photographs of observed silk with alignment keys (left) and bent status of silk with the radius of curvature of 2.5 mm (right). (c) Silk observed using 10× objective lens. Red dashed box indicates the observation area. (d) Raw images with different bending cycles (*i.e.*, before bending, 1st, 10th, and 100th bending). (e) Correlation coefficients of raw images with bending compared with the raw image before bending.

(Supplementary Figure 15: in the revised Supplementary Information)

Supplementary Figure 15. (a) Photograph of measurement setup for self-focusing effect under heating. (b) Temperature of silk material. Convective heating was used to raise the temperature. (c) Captured raw images with a time interval of 10 s. The temperature of silk was increased from 22.7 to 65.3 °C.

Comment #4: *The signal-to-noise ratio for the PUF readout should be mentioned.*

Our response to Comment #4: We are thankful for the reviewer's valuable comment. We agree that the signal-to-noise ratio (SNR) is a crucial factor for evaluating the performance of PUF. Hence, we conducted an SNR analysis for the LOP-PUF. In our system, thermal noise can be a strong readout noise. The SNR evaluation of the LOP-PUF requires quantitative thermal noises; thus, we introduced additive white Gaussian noise with a specific intensity into the image obtained by the LOP-PUF. Depending on the noise intensity, we estimated the robustness of the LOP-PUF to the noise.

Figure R5a presents the original image as well as the images treated by white Gaussian noise with different intensities. The SNR is defined as

$$SNR = 10 \times \log_{10} \left(\frac{Signal}{Noise} \right).$$

We applied different noise intensities to the original data for satisfying SNRs of 0, 2, 4, 6, 8, 10, and 12 dB.

In addition, we calculated the bit error rate (BER) using the noisy and original data to quantitatively highlight the robustness of our system to the noise. **Figure R5b** shows the bit acquisition process to simulate BER. In one process, we obtained 10-bit sequences with 64-bit sizes. Because the BER test requires a number of bit sequences, we repeated the bit acquisition process 100 times. **Figure R5c** shows BER of the LOP-PUF with different SNRs. The BER was acquired by comparing the bit maps from the noisy and original data. The LOP-PUF shows a low BER ($<10^{-4}$) over an SNR of 10 dB, *i.e.*, when the intensity of signal is 10 times higher those of noise.

Figure R5. (a) Original data and artificial images added with a noise of different SNRs (*i.e.*, 0, 2, 4, 6, 8, 10, and 12 dB). (b) Flow chart of the noise application process to the original data to estimate bit error rate (BER) of the LOP-PUF system. 10 bit sequences were repeated 100 times to perform acquisition and bit extraction. (c) BER by measuring the number of different bits compared to bits maps from the original data as a function of SNR.

Our revision to the manuscript: Same as Comment #3

Comment #5: *To reconcile a noisy response, the unreliability (that is, the bit error rate) of PUFs is a major hurdle for applications and should be shown.*

Our response to Comment #5: We thank you for the better suggestion. As suggested in *Comment #4*, we computationally studied the BER of the LOP-PUF. Nevertheless, this result cannot show the unreliability of the LOP-PUF, *i.e.*, bit errors in bit maps. Therefore, we also conducted bit generation under heated conditions. As we mentioned in *Comment #4*, thermal

noise is the strongest noise in our system. The readout system recognizes noisy signal as peak intensity, which is an undesired bit-seed. Thus, the response obtained by the LOP-PUF can be different depending on the temperature of image sensor.

Figure R6a shows the temperature variation of an image sensor in the readout system of a LOP-PUF module as a function of operating time with and without the blue light illumination with the center wavelength of 467 nm (optical intensity of $64 \mu\text{W}/\text{cm}^2$). Without an additional cooling system, the temperature of the image sensor increased more than $35 \text{ }^\circ\text{C}$ in 6 min, which causes the increase of the number and intensity of noise pixels (**Figure R2b**). In this temperature tendency, the number and intensity of noise pixels also increase, as shown in **Figure R6b**. Moreover, with the LED on, the temperature rose much faster, which can increase the production of noise signals even further (**Figure R6a**). We used commercial thermocouples to measure the temperatures.

Figure R6c presents the extracted bit responses. In the case of bit responses extracted under $35 \text{ }^\circ\text{C}$ and $36 \text{ }^\circ\text{C}$, the bit maps have noise bits, which are marked by red squares. This bit error situation can be addressed by using a simple cooling device (*i.e.*, a fan) for the image sensor.

Figure R6. (a) Temperature variation of an image sensor as a function of the operating time of a LOP-PUF module with and without under the blue light illumination with the center wavelength of 467 nm (optical intensity of $64 \mu\text{W}/\text{cm}^2$). (b) Maximum noise intensity (black solid line) and number of thermal noise (red solid line) as a function of the operating time of

a LOP-PUF module in dark. (c) Bit maps extracted from a LOP-PUF module at various temperatures of 25-36 °C.

To confirm the reduction of thermal noise, we installed a cooling fan under the image sensor (Figure R7a). Figure R7b presents the temperature of image sensor depending on the operation time of the LOP-PUF. The bit sequences by the LOP-PUF with the low temperature clearly demonstrate a reduced bit error (Figure R7c). Based on these results, we revised the manuscript to discuss the reliability of the LOP-PUF.

Figure R7. (a) Schematic of a LOP-PUF with a cooling system (i.e., fan). The cooling fan reduces a thermal noise in image sensor for stable bit extraction. (b) Temperature of image sensor as a function of operating time of the LOP-PUF with a cooling system. The

temperature of the image sensor was maintained at $\sim 27^\circ\text{C}$. (c) Bit sequences obtained by the LOP-PUF with the temperature control.

Our revision to the manuscript: Same as Comment #4

Comment #6: As far as I can see, the focal spots distribute sparsely in the obtained images (e.g., Figure 3I, Figure S6). This means there will be many “0” in digital keys, which will significantly reduce the encoding capacity of LOP-PUF. The authors should make a discussion on this issue. In the caption of Figure 3I, “four densities of silk fabric” should be “three densities”?

Our response to Comment #6: We are thankful for the reviewer’s valuable comment. As mentioned in the comment, the LOP-PUF generates sparse focal spots in raw images, resulting in a global bias of many “0” bits in digital keys. If a security key is biased with too many 0s or 1s, the actual coding capability is often diminished. To remove the bias of “0” bits, therefore, we applied an enhanced version of the von Neumann bias compression algorithm with two-pass tuple-output debiasing [R1]. Importantly, such debiasing process is useful not to comprise the actual coding capacity. After debiasing, our LOP-PUF has the nominal encoding capacity is 2^{768} , as responded in Comment #1. Furthermore, despite independent bit elements in bit sequences are considered, it has the high encoding capacity of 2^{345} ($\approx 7.1672 \times 10^{103}$).

For the mistake in the caption of Figure 3I, we have modified the typo in the revised manuscript.

References

[R1] Maes, R., van der Leest, V., van der Sluis, E. & Willems, F. Secure key generation from biased PUFs: Extended version. *J. Cryptographic Eng.* 6, 121–137 (2016).

Our revision to the manuscript:

(Figure 3I caption: in the revised main text)

“(I) Three-dimensional normalized intensities for **three** densities of silk fabric.”

Comment #7: Figure 1D, the picture of $d_{IS}=0$ seems to exhibit focal spots at the same sites as $d_{IS}=0.5$ mm, despite of a weaker intensity. While the upper circle in $d_{IS}=1.0$ mm does not show a focal spot for me. Can the authors explain it?

Our response to Comment #7: We sincerely thank you for bringing up this critical point that needs to be corrected in our manuscript. As mentioned, the picture of $d_{IS} = 0$ mm has high intensity spots at the same sites as $d_{IS} = 0.5$ mm. However, their maximum intensity is 192, which is relatively low compared to that of $d_{IS} = 0.5$ and 1 mm (**Figure R9**). Moreover, in the cut-off image, the white dots, which have higher intensity over cut-off value (*i.e.*, 80% of maximum intensity), are very small because the picture of $d_{IS} = 0$ mm has low contrast difference between high and low intensity spots.

For the upper circle in $d_{IS} = 1$ mm, the spot is clear when we observe the cut-off image (*i.e.*, binary image). However, a post-process for improving the visibility of focal spots negatively affects the visibility of the upper circle. Thus, we replaced the images in Figure 1D with the raw data of **Figure R9** in the revised manuscript.

Figure R9. (a) Highlighted images in Figure 1D. (b) Raw data, grayscale images, and cut-off images for each distance.

Our revision to the manuscript:
(Figure 1D: in the revised main text)

Figure 1. (D) Obtained image with a red LED at three different distances ($d_{IS} = 0$ mm, 0.5 mm, and 1.0 mm). The scale bar is 100 μ m.

(Line 2, Page 5: in the revised main text)

“Fig. 1D shows the experimental confirmation of the light concentration through random holes within the Fraunhofer region for d_{IS} (Supplementary Fig. 1)”

(Supplementary Figure 1: in the revised Supplementary Information)

Supplementary Figure 1. Raw data obtained from a LOP-PUF under the red-light illumination and its corresponding grayscale images and cut-off images for three different distances ($d_{IS} = 0$ mm, 0.5 mm, and 1.0 mm). For each distance, the maximum, minimum, and mean values are noted.

Comment #8: *In the reproducibility and uniqueness verification experiments, the sample volume of 16 silk ID cards is not sufficient to get reliable results. The authors should enlarge their sample capacity.*

Our response to Comment #8: We sincerely thank you for bringing up this critical point that should be corrected in our manuscript. Verification on a large number of samples is important for the demonstration of PUF characteristics. Thus, we increased the sample amount to 30 samples (**Figure R10**). Using the increased number of samples, we obtained intra-device Hamming distance, inter-device hamming distance (inter-device HD), and uniformity again (**Figure R11**). Moreover, we extracted bitmap from 30 silk ID cards via the LOP-PUF module for NIST SP-800 test (**Table R2**) and revised the manuscript accordingly.

Figure R10. (a) Photograph of the silk cards for bit extraction with the LOP-PUF module.

Figure R11. (a) Bit uniformity of the response bit data given by the ratio of ‘0’ and ‘1’ obtained by counting the number of digits at each silk ID card. (b) Inter-device HD and intra-device HD of each PUF response. (c) Magnified intra-device and inter-device HD for threshold of authentication. (d) 2D correlation of the LOP-PUF using 30 silk ID cards with 9 responses.

^a NIST test	statistical	^b p-value	Proportion	Result
Frequency		0.001873	133/135	Pass
Block frequency		0.573321	134/135	Pass
Cumulative sums		0.000730, 0.021542	133/135, 134/135	Pass
Runs		0.382509	134/135	Pass
Longest run		0.998338	135/135	Pass
Serial		0.369488, 0.320255	132/135, 134/135	Pass
Approximate entropy		0.573321	134/135	Pass

Table R2. Summary of the randomness tests of binary sequences generated from the LOP-PUF using silk. ^a NIST tests are performed using 135 sequences of 128 bits each such that 17,280 bits (*i.e.*, digitized keys) collected from 30 different PUFs are tested. The chi-squared (χ^2) distribution is used to compare the goodness-of-fit of the p-value distribution of the blocks from the entire bitstream to the expected distribution. The bitstream is considered to be random only if the p-value ≥ 0.0001 . ^b If the pass rate exceeds the minimum rate ($>130/135$) for each test, it is considered as a pass.

Our revision to the manuscript:
 (Figure 5: in the revised main text)

Figure 5. Generated bits and features of the LOP-PUF. (A) Fifteen bit sequences obtained by the LOP-PUF for three spectrally separate LEDs. The three LEDs were positioned at an angle in the LOP-PUF to illuminate the light obliquely. Total of the obtained bit sequences (i.e., 1–30 of ID cards) are shown in Supplementary Fig. 18. (B) Bit uniformity of the response bit data given by the ratio of ‘0’ and ‘1’ obtained by counting the number of digits for each silk ID card. (C) Inter-device HD and intra-device HD of each PUF response. (D) Magnified intra-device and inter-device HD for threshold of authentication. (E) 2D correlation of the LOP-PUF using 30 silk ID cards with 9 responses. The black solid lines delineate each silk ID card and the color bars indicate the colors of the LEDs.

(Table 1: in the revised main text)

^a NIST statistical test	^b p-value	Proportion	Result
Frequency	0.001873	133/135	Pass
Block frequency	0.573321	134/135	Pass
Cumulative sums	0.000730, 0.021542	133/135, 134/135	Pass
Runs	0.382509	134/135	Pass
Longest run	0.998338	135/135	Pass
Serial	0.369488, 0.320255	132/135, 134/135	Pass
Approximate entropy	0.573321	134/135	Pass

Table 1. Summary of the randomness tests of binary sequences generated from the LOP-PUFs. ^aNIST tests are performed using 135 sequences of 128 bits each such that 17,280 bits (*i.e.*, digitized keys) collected from 30 different PUFs are tested. The chi-squared (χ^2) distribution is used to compare the goodness-of-fit of the p-value distribution of the blocks from the entire bitstream to the expected distribution. The bitstream is considered to be random only if the p-value ≥ 0.0001 . ^bIf the pass rate exceeded the minimum rate ($>130/135$) for each test, it was considered to be a pass.

(Supplementary Figure 16: in the revised Supplementary Information)

Supplementary Figure 16. The bit sequences from 30 different silk ID cards obtained by the LOP-PUF for three spectrally separate LEDs. The three LEDs were positioned at an angle in the LOP-PUF to illuminate the light obliquely.

(Supplementary Figure 17: in the revised Supplementary Information)

B

Supplementary Figure 17. (B) Photography of the silk cards for bit extraction with the LOP-PUF module.

(Supplementary Figure 20: in the revised Supplementary Information)

Supplementary Figure 20. Bitmap extracted from 30 silk ID cards via a LOP-PUF module through nine challenge–response pairs (*i.e.*, red +15°, green +15°, blue +15°, red 0°, green 0°, blue 0°, red –15°, green –15°, and blue –15°). A stream of 64 bits was generated by each challenge–response pair. As a result, a total of 17,280 bits were collected for the NIST randomness test.

Comment #9: The authors used HD to represent the reproducibility and uniqueness of LOP-PUF, but detailed description on HD calculation is absent.

Our response to Comment #9: Thank you for the helpful comment. We have added the following equations for the definition of Hamming distance (HD), inter-device HD, and intra-device HD in the revised manuscript [R1].

The bit uniformity is defined as

$$\text{Hamming distance} = \frac{\#(K_{i,l} \neq K_{j,l})}{s}, \quad (2)$$

where K_l is the l th binary bit of the key and s is the key size.

$$\text{Inter-device HD} = \frac{2}{q(q-1)} \sum_{i=1}^{q-1} \sum_{j=i+1}^q \frac{HD(K_i, K_j)}{2}, \quad (3)$$

where K_i and K_j are s -bit keys of the i^{th} PUF device and the j^{th} PUF device among q different PUFs, respectively.

$$\text{Intra-device HD} = \frac{1}{m} \sum_{i=1}^m \frac{HD(K_i, K_{i,t})}{2}, \quad (4)$$

where $K_{i,t}$ represents the s -bit keys of the i^{th} PUF device at t^{th} time among m different acquisition times, respectively. We have revised the manuscript to reflect the above equations.

References

[R1] MAITI, Abhranil; GUNREDDY, Vikash; SCHAUMONT, Patrick. A systematic method to evaluate and compare the performance of physical unclonable functions. In: *Embedded systems design with FPGAs*. Springer, New York, NY, (2013). p. 245-267.

Our revision to the manuscript:

(Line 19, Page 11: in the revised main text)

“The bit uniformity is defined as

$$\text{Bit uniformity} = \frac{1}{s} \sum_{l=1}^s K_l, \quad (5)$$

where K_l is the l^{th} binary bit of the key and s is the key size. Each bit sequence was extracted with a von Neumann extractor, which enhanced the bit uniformity up to 0.4972 (Fig. 5B).

The reproducibility of the PUF responses was confirmed by calculating the intra-device HD when the same challenge was applied to the same PUF-tag.

$$\text{Hamming distance (HD)} = \frac{\#(K_{i,l} \neq K_{j,l})}{s}, \quad (6)$$

$$\text{Intra-device HD} = \frac{1}{m} \sum_{i=1}^m \frac{HD(K_i, K_{i,t})}{2}, \quad (3)$$

where $K_{i,t}$ represents the s -bit keys of the i^{th} PUF device at t^{th} time among m different acquisition times. The uniqueness of the PUFs was verified by evaluating the inter-device HD, which showed the difference in the bitstreams between two different PUFs.

$$\text{Inter-device HD} = \frac{2}{q(q-1)} \sum_{i=1}^{q-1} \sum_{j=i+1}^q \frac{HD(K_i, K_j)}{2}, \quad (4)$$

where $K_{i,l}$ is the l^{th} binary bit of the key in the i^{th} PUF device, $K_{j,l}$ is the l^{th} binary bit of the key in the j^{th} PUF device, and s is the size of the key. In an ideal PUF system, the intra-device HD is 0 for stable acquisition and the inter-device HD is 0.5 for uniqueness. The estimated intra-device HD exhibited a relatively low mean value of 0.0309 for 30 different silk ID cards over the same 10 challenge cycles. In addition, the inter-device HD was estimated from 30 silk ID cards over 9 challenges and had a mean value of 0.4990 (Fig. 5C).”

Comment #10: The authors did not give detailed descriptions on how to get the threshold for PUF digitization.

Our response to Comment #10: We appreciate the reviewer for the fruitful comment that improves the quality of the manuscript. We removed less than 80% of the intensity distribution to threshold the raw data obtained with LOP-PUF. Strong peaks were observed in the raw data, as shown in **Figure R12a**. For binarization, we observed the histogram of the pixel intensity to determine the threshold value. The dip is observed in the full range indicated in the histogram, which is ~80% (*i.e.*, 56) of the maximum value (**Figure R12b**). **Figure R12c** shows the obtained binary data by applying these thresholds.

Figure R12. (a) Grayscale image of raw data obtained using the LOP-PUF. (b) Histogram of the grayscale image that shows a dip at ~80% of the intensity range. (c) Binary image obtained using 80% of threshold.

Additionally, we theoretically estimated the authentication threshold of the LOP-PUF by using intra-device hamming distance (Intra-HD) and inter-device hamming distance (Inter-device HD). To confirm the threshold of PUF, we calculated intra-device HD and inter-device HD as shown in **Figure R13a**. To estimate the authentication threshold,

Gaussian fitting was used to determine the overlapped area (**Figure R13b**). The false negative rate is the probability that an initially authenticated PUF will fail authentication. The cut-off threshold (HD = 0.225) returns a false positive rate of 3.1728×10^{-24} and a false negative rate of 4.7417×10^{-24} . The result with a threshold value of 0.225 indicates a high authentication practicality with a relatively very small value of 4.7417×10^{-24} as false negative rate. We have added a detailed description of these results in the revised manuscript.

Figure R13. (a) Intra-device HD, and inter-device HD with the LOP-PUF. (b) Magnified graph of intra-device HD and inter-device HD. The overlapped areas present false negative (blue diagonal pattern area) and false positive (red diagonal pattern area).

Our revision to the manuscript:

(Line 9, Page 13: in the revised main text)

“With calculated intra-device HD, and inter-device HD, we estimated the threshold of authentication mode. To estimate authentication threshold, Gaussian fitting was used to determine the overlapped area (Fig. 5D). The false negative rate is the probability that an initially authenticated PUF will fail authentication with 0.225 of authentication threshold.”

Comment #11: *In Method section, some SI Figure number is wrong. For example, line 315, “Fig. S13” should actually be Fig. S14. Please check the whole manuscript to ensure consistency.*

Our response to Comment #11: We apologize for this oversight. We have now checked the order and number of figures in the whole manuscript again, especially since we have considerably updated the manuscript in this round of revisions, wherein the number of figures has changed.

Thank you, once again, for your insightful comments. We feel that they have helped improve the quality of the manuscript significantly.

Reviewer #2:

***Summary Comments:** This paper develops a lens-free, optical, and portable PUF based on stochastically manifested diffraction using native silk fibers. By optimizing the distance between a silk PUF-tag and an image sensor, this simple apparatus engineering easily formed random light-spot patterns with high-intensity contrast. Overall, this work is interesting and novel. Most of the paragraphs are clearly written. Here I only have a major concern about the PUF performance. For example, the temperature dependent CRPs and the power consumption are missing. Please refer to my detail suggestions:*

Our response to summary comments: We sincerely appreciate your valuable comments, which were helpful in improving the quality of our manuscript significantly. We have revised our manuscript accordingly and our responses are described below.

***Comment #1:** A high quality PUF has also to be reliable. This means that at any point in time, under any operation conditions, the CRPs have to be unchanged. Although the authors have estimated the intra-device HD to be 0.1282, the temperature may induce changes of the width of the stochastic randomly distributed holes produced by silk fibers, which may further result in changes of the generated bitstreams. In their system, how large is the sensitivity to temperature.*

Our response to Comment #1: Thank you for the helpful comment. We established a measurement setup to investigate the temperature dependency of silk (**Figure R1a**). In this measurement, we heated the silk using a convective heater (hair dryer) and a thermocouple logged the temperature of silk. **Figure R1b** presents the measured temperature of silk under convective heating. While heating the silk, we captured the self-focusing effect of silk at a time interval of 10 s (**Figure R1c**). These captured images display the invariant self-focal spots. Our system has a temperature dependency due to thermal noise of image sensor; however, the silk material does not vary near or over the operating temperature of the module.

Figure R1. (a) Photograph of measurement setup for self-focusing effect under heating. (b) Temperature of silk material. Convective heating was used to raise the temperature. (c) Captured raw images with a time interval of 10 s. The temperature of silk was increased from 22.7 to 65.3 °C.

However, an image sensor is usually sensitive to the temperature. In this case, the readout system of a LOP-PUF module recognizes a noisy signal as a peak intensity, which is the undesirable bit seed. **Figure R2a** shows the temperature variation of an image sensor in the readout system of a LOP-PUF module as a function of operating time with and without the blue light illumination with the center wavelength of 467 nm (optical intensity of $64 \mu\text{W}/\text{cm}^2$). Without an additional cooling system, the temperature of the image sensor increased more than 35 °C after six minutes, which causes the increase of the number and intensity of noise pixels (**Figure R2b**). Moreover, under light illumination (i.e., LED on-state), the temperature of the image sensor rose much faster, resulting in a possibly larger number of noise signals (**Figure R2a**). **Figure R2c** presents the extracted bit sequences at different temperatures. At high temperatures of 35 °C and 36 °C, there are discrepancy on the bit maps compared to ones at lower temperatures, which are marked by red squares. This is attributable to the thermal noises. However, such a bit error situation can be solved by applying a simple cooling device (i.e., a fan) to the image sensor.

Figure R2. (a) Temperature variation of an image sensor as a function of the operating time of a LOP-PUF module with and without under the blue light illumination with the center wavelength of 467 nm (optical intensity of $64 \mu\text{W}/\text{cm}^2$). (b) Maximum noise intensity (black solid line) and number of thermal noise (red solid line) as a function of the operating time of a LOP-PUF module in dark. (c) Bit maps extracted from a LOP-PUF module at various temperatures of 25-36 °C.

To confirm the reduction of thermal noise, we installed a cooling fan under the image sensor (**Figure R3a**). **Figure R3b** presents the temperature of image sensor depending on the operation time of the LOP-PUF. The bit sequences by the LOP-PUF with the low temperature clearly demonstrate a reduced bit error (**Figure R3c**). Based on these results, we included new figure to discuss the reliability of the LOP-PUF in the revised manuscript.

Figure R3. (a) Schematic of a LOP-PUF with a cooling system. The cooling fan reduces thermal noise in the image sensor for stable bit extraction. (b) Temperature of image sensor as a function of operating time of the LOP-PUF with a cooling system. The temperature of the image sensor was maintained at $\sim 27^{\circ}\text{C}$. (c) Bit sequences obtained by the LOP-PUF with the temperature control.

Our revision to the manuscript:

(Line 15, Page 9: in the revised main text)

“Reliability of LOP-PUFs

A high-quality PUF should have high reliability against external and internal factors such as thermal noise and aging issues. First, the robustness of the LOP-PUF to the thermal noise was estimated by simulating bit error rate (BER) depending on the signal-to-noise ratio (SNR). Fig. 4A illustrates the bit acquisition process to simulate the BER of the LOP-PUF. Ten-bit sequences with 64-bit sizes are obtained in one acquisition process, and this process was repeated a hundred times to acquire a thousand bit sequences because the BER test demands a number of bit sequences. Fig. 4B presents the original data as well as the data treated by white Gaussian noise with different intensities to satisfy the SNRs of 0, 6, and 12 dB. Other SNRs of treated data (2, 4, 8, 10 dB) are shown in Supplementary Fig. 9. Fig. 4C shows the BER of the LOP-PUF versus SNRs. The LOP-PUF shows a low BER ($<10^{-4}$) over the SNR of 10 dB, *i.e.*, when the intensity of signal is 10 times higher than that of noise.

To investigate the actual influence of thermal noise in the LOP-PUF, continuous operations of the LOP-PUF were conducted by measuring the temperature of the module and capturing the responses without and with temperature control (Figs. 4D and 4E). The obtained responses without and with temperature control are displayed in Supplementary Figs. 10 and 11, respectively. Without the temperature control, the image sensor of LOP-PUF heated up to ~ 35 °C in 6 min; however, with a cooling fan, the temperature maintained at ~ 27 °C throughout (Fig. 4D). The LOP-PUF without a cooling fan tends to cause error bits during the operation (Fig. 4E). This result demonstrates that the LOP-PUF without the cooling fan shows reduced correction rate (*i.e.*, # Correct bits / # Total bits) to 90% as the operating time increases (*i.e.*, increase in temperature of image sensor). In contrast, the cooling fan ensures that the LOP-PUF shows a stable correction rate of $\sim 100\%$.

In addition, because the LOP-PUF adopts silk material as the PUF-tag, biodegradability is an important issue as it may degrade the reliability of the LOP-PUF. A customized measurement setup was used to explore the biodegradability effect of silk (Fig. 4F). Because silk material is susceptible to humidity, our setup could control the humidity using silica gel and humidifier. The initial relative humidity (RH) was controlled at a low value, *i.e.*, 30.2%, and then it is increased up to 59.7% using the humidifier. While controlling the RH, the LOP-PUF captured the responses and the obtained responses were transformed to bit sequences. The bit extraction was conducted 10 times to confirm the stability of bit sequences.

The extracted bits at the RH of 30.2, 34.8, 39.8, 44.8, and 50.0% show robust results (*i.e.*, zero-bit error). However, the extracted bits at the RH of 55.2 and 59.7 % exhibit 2- and 8-bit errors, respectively, during 10 times generation (Fig. 4G and Supplementary Fig. 12). During the humidity test, the LOP-PUF exhibits an error rate (*i.e.*, # Error bits / # Total bits) of 0.0022. Additionally, a long-term measurement was performed for a week under the conditions of room temperature and humidity. In this result, an error of 1 bit, which corresponds to an error rate of 0.0004, appeared at the sixth day during the seven days (Fig.

4H and Supplementary Fig. 13). Moreover, the native silks show good flexibility against mechanical and thermal stress (Supplementary Figs. 14 and 15). These results imply that the utilization of the LOP-PUF in highly humid environments requires the use of non-biodegradable fibrous medium made by water-resistant polymers; however, for indoor environments, the silk material can serve as PUF-tag for a long time.

”

(Figure 4: in the revised main text)

Figure 4. Reliability test of the LOP-PUF. (A) Flow chart of signal-to-noise rate (SNR) obtained by introducing additive white Gaussian noise (AWGN) computationally. (B) Original data obtained under the blue light illumination with the center wavelength of 467 nm and treated data with AWGN to satisfy SNRs of 0, 6, and 12 dB. (C) Bit error rate according to SNR. (D) Schematic of the LOP-PUF module with a cooling fan, which can reduce thermal noise. (E) (top) Bit responses of a LOP-PUF without a cooling fan at various temperature conditions (*i.e.*, 30, 31, 32, 33, 34, 35, and 36 °C). (bottom) Ratio of number of corrected and error bits and temperature variation versus operating time of the LOP-PUF module with and without a cooling fan. The red line and blue line represent the data without

and with a cooling fan, respectively, according to operation time. (F) Schematic of measurement setup for humidity control. The LOP-PUF module is placed in the enclosed chamber with nebulizer, hygrometer, and silica gel. The silica gel is used to set the initial humidity. (G) Number of correct bits and error bits obtained from the LOP-PUF at different relative humidity conditions (*i.e.*, 30.2, 34.8, 39.8, 44.8, 50.0, 55.2, and 59.7 %). (H) Number of correct and error bits obtained from the LOP-PUF over seven days under room temperature and ~30% relative humidity.

(Supplementary Figure 9: in the revised Supplementary Information)

Supplementary Figure 9. Original data and artificial images added with a noise of different SNRs (*i.e.*, 0, 2, 4, 6, 8, 10, and 12 dB).

(Supplementary Figure 10: in the revised Supplementary Information)

Supplementary Figure 10. (a) Temperature variation of an image sensor as a function of the operating time of a LOP-PUF module with and without under the blue light illumination with the center wavelength of 467 nm (optical intensity of $64 \mu\text{W}/\text{cm}^2$). (b) Maximum noise intensity (black solid line) and number of thermal noise (red solid line) as a function of the operating time of a LOP-PUF module in dark. (c) Bit maps extracted from a LOP-PUF module at various temperatures of 25-36 °C.

(Supplementary Figure 11: in the revised Supplementary Information)

Supplementary Figure 11. Bit sequences obtained by a LOP-PUF with the temperature control (*i.e.*, 27 °C).

(Supplementary Figure 12: in the revised Supplementary Information)

Supplementary Figure 12. (a, b) Schematic (a) and photographs (b) of measurement setup for various relative humidity conditions. Silica gels were used to lower the initial humidity, and the humidifier raised the humidity. (c) The obtained responses at each relative humidity value under the blue light illumination with the center wavelength of 467 nm (*i.e.*, RH = 30.2, 34.8, 39.8, 44.8, 50.0, 55.2, and 59.7 %).

(Supplementary Figure 13: in the revised Supplementary Information)

Supplementary Figure 13. Raw data (left) and bit response (right) obtained with the LOP-PUF over seven days. The bit responses show '1' bit error on the sixth day in the measurement conducted over a seven-day period. Data acquisition was repeated 10 times per measurement under the blue light illumination with the center wavelength of 467 nm.

(Supplementary Figure 14: in the revised Supplementary Information)

Supplementary Figure 14. (a) Schematic illustration of measurement setup for bending test. (b) Photographs of observed silk with alignment keys (left) and bent status of silk with the radius of curvature of 2.5 mm (right). (c) Silk observed using 10× objective lens. Red dashed box indicates the observation area. (d) Raw images with different bending cycles (*i.e.*, before bending, 1st, 10th, and 100th bending). (e) Correlation coefficients of raw images with bending compared with the raw image before bending.

(Supplementary Figure 15: in the revised Supplementary Information)

Supplementary Figure 15. (a) Photograph of measurement setup for self-focusing effect under heating. (b) Temperature of silk material. Convective heating was used to raise the temperature. (c) Captured raw images with a time interval of 10 s. The temperature of silk was increased from 22.7 to 65.3 °C.

Comment #2: This work also lacks important information such as power consumption, speed and operating voltage of implemented silk fiber-based optical PUF.

Our response to Comment #2: We have now addressed this issue. First, we measured the power consumption of our module composed of lighting part (*i.e.*, three LEDs) and reading part (*i.e.*, image sensor). Because our module is connected with a laptop for the operation of LEDs and data acquisition of image sensor, we added two power meters between the PUF module and laptop (**Figure R4a**). The operating voltage was set to 5 V for both lighting and reading parts. The logged currents and powers are shown in the top of **Figure R4b**. We additionally calculated the working time of our module when using a commercial battery with a capacity of 7500 mAh. This calculation shows that our module can continuously operate over a day (~27 h).

Figure R4. (a) Schematic for power consumption measurement of the LOP-PUF. Two USB-type power meters were used to measure the power consumption. (b) (top) Measured current and power and (bottom) working time of LOP-PUF operated by a portable battery with a capacity of 7500 mAh. The operating voltage was set to 5 V.

The operating speed is determined by the data acquisition part as the LEDs always turn on during the time period of module working. When we use programmable languages (*i.e.*, Python and Matlab), the frame rate per second (FPS) was 7. When considering all data processing, the LOP-PUF can process bit transformation from the random seed images with 0.422 s (*i.e.*, 0.194 s of median filter, 0.109 s of Gaussian filter, 0.053 s of area filter, 0.019 s of von Neumann debiasing, and etc.). Therefore, the working speed is ~ 2.3 FPS. We have now explained this in the revised manuscript.

Our revision to the manuscript:

(Line 11, Page 13: in the revised main text)

“Seven statistical evaluations for randomness were performed and all tests were successfully passed (Table 1). **The LOP-PUF can process bit transformation from random seed images with 0.422 s for each data processing. Additionally, the LOP-PUF requires low power**

consumption: the working time is estimated at ~27 h when using a 7500 mAh portable battery (Supplementary Fig. 21).”

(Figure S21: in the revised Supplementary Information)

Supplementary Figure 21. (a) Schematic of power consumption measurement for the LOP-PUF. Two USB-type power meters were used to measure the power consumption. (b) (top) Measured current and power and (bottom) working time of LOP-PUF operated by a portable battery with a capacity of 7500 mAh. The operating voltage was set to 5 V.

Comment #3: *It would be better to cite latest works as introducing a certain technology. For example, when authors mentioned various mechanism-based PUFs, magnetics PUF, very recently, was reported using the methods of spin-orbit-torque driven magnetic switching (Journal of Applied Physics 128, 033904 (2020), IEEE Electron Device Letters 42, 597 (2021)...) or interface anisotropy modulation (18, 7211(2018))*

Our response to Comment #3: As requested, we have now discussed the latest works in the introduction part.

Our revision to the manuscript:

(Line 26, Page 20: in the revised main text)

“Marukame, T., Tanamoto, T. & Mitani, Y. Extracting physically unclonable function from spin transfer switching characteristics in magnetic tunnel junctions. *IEEE T. Magn.* **50**, 1–4 (2014). doi:10.1109/TMAG.2014.2325646.

21. Chen, H. *et al.* Highly Secure Physically Unclonable Cryptographic Primitives Based on Interfacial Magnetic Anisotropy. *Nano Lett.* **18**, 7211–7216 (2018). doi:10.1021/acs.nanolett.8b03338
22. Finocchio, G. *et al.* Spin-orbit torque based physical unclonable function. *J. Appl. Phys.* **128**, 033904 (2020). doi:10.1063/5.0013408
23. Cao, Z. *et al.* Reconfigurable Physical Unclonable Function Based on Spin-Orbit Torque Induced Chiral Domain Wall Motion. *IEEE Electron Device Lett.* **42**, 597–600 (2021). doi: 10.1109/LED.2021.3057638

24. Lee, J. W. et al. A technique to build a secret key in integrated circuits for identification and authentication applications. In: 2004 Symposium on VLSI Circuits. Digest of Technical Papers (IEEE Cat. No. 04CH37525) 176–179 (IEEE, 2004).”

Thank you, once again, for your insightful comments. We feel that they have helped improve the quality of the manuscript significantly.

Other minor modifications

1. The order and number of figures were changed during the revision process.
2. We added acknowledgements for funding source as below.

Our revision to the manuscript:

(Acknowledgements: in the revised main text)

Acknowledgments

This research was supported by the National Research Foundation of Korea (NRF-2020R1A2C2004983/2018M3D1A1058997) and by the GIST Research Institute (GRI) grant funded by the GIST in 2021. This work was also supported by the Institute of Information & Communications Technology Planning & Evaluation (IITP) through a grant funded by the Korean government (MSIT) (No.2020-0-01000, Light field and LiDAR sensor fusion systems for full self-driving).

3. Minor typos were corrected.

4. The section of ‘Data availability’ was added in the revised manuscript for readership.

Our revision to the manuscript:

(Data availability: in the revised main text)

Data availability

The data and code supporting the findings of this study are available from the corresponding authors upon reasonable request. The source data in Fig. 4 and Supplementary Figs. 16 and 20 are provided as source data files.

5. The affiliation for one of authors was changed as below.

Our revision to the manuscript:

(Affiliations: in the revised main text)

“Min Seok Kim^{†1}, Gil Ju Lee^{†1,2}, Jung Woo Leem³, Seungho Choi⁴, Young L. Kim^{3,5*}, and Young Min Song^{1,6,7*}

¹*School of Electrical Engineering and Computer Science (EECS), Gwangju Institute of Science and Technology, 123, Cheomdangwagi-ro, Buk-gu, Gwangju, Republic of Korea, 61005*

²*Department of Electronics Engineering, Pusan National University, 2 Busandaehakro 63 beon-gil, Geumjeong-gu, Busan 46241, Republic of Korea*

³*Weldon School of Biomedical Engineering, Purdue University, West Lafayette, Indiana 47907, United States*

⁴*Department of Biomedical Engineering, Yonsei University, Wonju 220-710, Republic of Korea*

⁵*Purdue Quantum Science and Engineering Institute, West Lafayette, Indiana 47907, United States*

⁶*Anti-Viral Research Center, Gwangju Institute of Science and Technology (GIST), 123, Cheomdangwagi-ro, Bukgu, Gwangju, Republic of Korea, 61005*

⁷*AI Graduate School, Gwangju Institute of Science and Technology (GIST), 123, Cheomdangwagi-ro, Bukgu, Gwangju, Republic of Korea, 61005”*

REVIEWERS' COMMENTS

Reviewer #1 (Remarks to the Author):

In general, the authors have greatly improved the quality in this version. I don't have comments except a suggestion to cite below two articles when the authors discussed the encoding capacity of optical PUFs.

(1) Matter, 2020, 3, 2160-2180; (2) Nature Communications, 2020, 11, 516.

Reviewer #2 (Remarks to the Author):

The authors have addressed my concerns in the response to comments and the revised manuscript. I recommend it for publication in Nature Communications.

Reviewer #1:

Comment #1: In general, the authors have greatly improved the quality in this version. I don't have comments except a suggestion to cite below two articles when the authors discussed the encoding capacity of optical PUFs.

(1) Matter, 2020, 3, 2160-2180; (2) Nature Communications, 2020, 11, 516.

Our response to Comment #1: As suggested, we have now discussed the latest works for encoding capacity of optical PUFs.

Our revision to the manuscript:

(Line 16, Page 12: in the revised main text)

“Moreover, the encoding capability of our LOP-PUF was estimated theoretically and practically. The conventional PUF systems can have an encoding capacity of c^s , where c is bit states and s is the size of bit sequence^[36,37]. Our LOP-PUF has a c value of 2 and an s value of 768; hence, the theoretical encoding capacity is 2^{768} . However, many PUF systems have estimated their encoding capacity by considering independent bit elements in bit sequences, which is called the ‘degree of freedom’ (DoF):”

(Reference: in the revised main text)

“35. Shi, N. N. et al. Nanostructured fibers as a versatile photonic platform: Radiative cooling and waveguiding through transverse Anderson localization. *Light Sci. Appl.* **7**, 37 (2018). doi:10.1038/s41377-018-0033-x.

36. Gu, Y. et al. Gap-enhanced Raman tags for physically unclonable anticounterfeiting labels. *Nat. Commun.* **11**, 1–13 (2020). doi:10.1038/s41467-019-14070-9

37. Jing, L. et al. Multigenerational Crumpling of 2D Materials for Anticounterfeiting Patterns with Deep Learning Authentication. *Matter* **3**, 2160–2180 (2020). doi: 10.1016/j.matt.2020.10.005

38. Rukhin, A., et al. A statistical test suite for random and pseudorandom number generators for cryptographic applications. Vol. National Institute of Standards and Technology (NIST), 800–822 (NIST Special Publication, Gaithersburg, 2010).

”

Thank you, once again, for your insightful comments. We feel that they have helped improve the quality of the manuscript significantly.

Reviewer #2:

Comment #1: The authors have addressed my concerns in the response to comments and the revised manuscript. I recommend it for publication in Nature Communications.

Our response to Comment #1: We are grateful for the enthusiastic support from Reviewer #2.

Thank you, once again, for your insightful comments. We feel that they have helped improve the quality of the manuscript significantly.

Other minor modifications

1. The title has been slightly modified for the clarity of meaning.

(Title : in the revised main text)

Revisiting silk: a lens-free optical **physical** unclonable function

(Title : in the revised supplementary text)

Revisiting silk: a lens-free optical **physical** unclonable function

2. We added acknowledgements for funding source as below.

Our revision to the manuscript:

(Acknowledgements: in the revised main text)

Acknowledgments

This research was supported by the National Research Foundation of Korea (NRF-2020R1A2C2004983/2018M3D1A1058997) and by the GIST Research Institute (GRI) grant funded by the GIST in 2021. This work was also supported by the Institute of Information & Communications Technology Planning & Evaluation (IITP) through a grant funded by the Korean government (MSIT) (No.2020-0-01000, Light field and LiDAR sensor fusion systems for full self-driving) and by the United States Air Force Office of Scientific Research (FA2386-17-1-4072).

3. We added Author contributions

Our revision to the manuscript:

(Author contributions: in the revised main text)

Author contributions

M.S.K. and G.J.L. developed the the idea of this study. M.S.K. and G.J.L. worked on the device fabrication, the optical measurements, and the key generation. J.W.L., S.H.C., Y.L.K. and Y.M.S. participated in the analyses. M.S.K. and G.J.L. wrote the paper. Y.L.K. and Y.M.S. directed the overall research. All authors contributed to writing the manuscript.

4. We added Competing interests section

Our revision to the manuscript:

(Competing interests: in the revised main text)

Competing interests

The authors declare no competing interests.

5. The section of ‘Data availability’ was changed in the revised manuscript for readership.

Our revision to the manuscript:

(Data availability: in the revised main text)

Data availability

The image data are available in a github repository at https://github.com/seok9643/Silk_PUF. The source data in Fig. 4 and Supplementary Figs. 16 and 20 are provided as source data files. The extra data and code supporting the findings of this study are available from the corresponding authors upon reasonable request.

5. We changed the image used in figure 6d-e for encoding and decoding to avoid copyright ambiguity of the image. Figure 6f has also been updated with data,

Our revision to the manuscript:

(Figure 6: in the revised main text)

Fig. 6 Applications of the LOP-PUF: authentication and data encoding/decoding. (a) Schematic illustration for authentication. ‘Enrollment’ is the step where the manufacturer registers the PUF keys from silk ID cards in the data center. ‘Authentication’ is the process for

the validation of unknown users using stored PUF keys. (b) An illustration of a Brute Force attack test for estimating the defense ability of the authentication system by substituting a random key for fake authentication. The average brake time, B_{time} , was averaged using the individual B_{time} values for ten attempts. (c) A graph of the breaking time using the Brute Force attack in our proposed system as a function of the key size. Using three-color space allowed the creation of a large number of key sizes for a strong security level. (d) Description for the application of data encryption in the LOP-PUF. Successful data encoding and decoding with encryption were enabled only when software and hardware binding was performed. With or without PUF operating, input data was stored as encrypted or vulnerable data, and the stored data was either decoded properly or illegible. The purple and green boxes indicate success or failure of the data restoration and encryption depending on the presence of the PUF operation. The letters ‘S’ and ‘F’ mean success and fail, respectively. (e) Data encryption and decryption in the pictured materials using five responses with different ‘1’ prediction ratios of 10, 30, 50, 70, and 90%. The PUF with a half-prediction ratio was generated by the LOP-PUF using silk. (f) HD of the OD-ED and OD-DD. The red solid lines refer to the distributions of the HD of the OD-ED.

(Supplementary Figure 22: in the revised Supplementary Information)

Supplementary Fig. 22 **Illustration of the data encoding and decoding system.** The original data consisted of three-color space. For encoding and decoding, an XOR operation was performed on each channel data with the extracted PUF key data.